# Blended Matching Pursuit

**Cyrille W. Combettes**
Georgia Institute of Technology
Atlanta, GA, USA
`cyrille@gatech.edu`

**Sebastian Pokutta**
Zuse Institute Berlin and TU Berlin
Berlin, Germany
`pokutta@zib.de`

## Abstract

Matching pursuit algorithms are an important class of algorithms in signal processing and machine learning. We present a *blended matching pursuit* algorithm, combining coordinate descent-like steps with stronger gradient descent steps, for minimizing a smooth convex function over a linear space spanned by a set of atoms. We derive sublinear to linear convergence rates according to the smoothness and sharpness orders of the function and demonstrate computational superiority of our approach. In particular, we derive linear rates for a large class of non-strongly convex functions, and we demonstrate in experiments that our algorithm enjoys very fast rates of convergence and wall-clock speed while maintaining a sparsity of iterates very comparable to that of the (much slower) orthogonal matching pursuit.

## 1 Introduction

Let $\mathcal{H}$ be a separable real Hilbert space, $\mathcal{D} \subset \mathcal{H}$ be a dictionary, and $f : \mathcal{H} \to \mathbb{R}$ be a smooth convex function. In this paper, we aim at solving the problem:

$$\text{Find a solution to } \min_{x \in \mathcal{H}} f(x) \text{ which is sparse relative to } \mathcal{D}. \tag{1}$$

Together with fast convergence, achieving high sparsity, i.e., keeping the iterates as linear combinations of a *small* number of atoms in the dictionary $\mathcal{D}$, is a primary objective and leads to better generalization, interpretability, and decision-making in machine learning. In signal processing, Problem (1) encompasses a wide range of applications, including compressed sensing, signal denoising, and information retrieval, and is often solved with the Matching Pursuit algorithm [Mallat and Zhang, 1993]. Our approach is inspired by the Blended Conditional Gradients algorithm [Braun et al., 2019] which solves the constrained setting of Problem (1), i.e., minimizing $f$ over the convex hull $\text{conv}(\mathcal{D})$ of the dictionary, and is ultimately based on the Frank-Wolfe algorithm [Frank and Wolfe, 1956] a.k.a. Conditional Gradient algorithm [Levitin and Polyak, 1966]. It enhances the vanilla Frank-Wolfe algorithm by replacing the linear minimization oracle with a *weak-separation oracle* [Braun et al., 2017] and by blending the traditional Frank-Wolfe steps with *lazified* Frank-Wolfe steps and projected gradient steps, while still avoiding projections. Frank-Wolfe algorithms are particularly well-suited for problems with a desired sparsity in the solution (see, e.g., Jaggi [2013] and the references therein) however, from an optimization perspective, although they approximate the optimal descent direction $-\nabla f(x_t)$ via the linear minimization oracle $v_t^{\text{FW}} \leftarrow \arg\min_{\mathcal{D}} \langle \nabla f(x_t), v \rangle$, they move in the direction $v_t^{\text{FW}} - x_t$ in order to ensure feasibility, which provides less progress.

An analogy between Frank-Wolfe algorithms and the unconstrained Problem (1) was proposed by Locatello et al. [2017]. They unified the Frank-Wolfe and Matching Pursuit algorithms, and proposed a Generalized Matching Pursuit algorithm (GMP) and an Orthogonal Matching Pursuit algorithm (OMP) for solving Problem (1), which descend in the directions $v_t^{\text{FW}}$. Essentially, Locatello et al. [2017] established that GMP corresponds to the vanilla Frank-Wolfe algorithm and OMP corresponds to the Fully-Corrective Frank-Wolfe algorithm. GMP and OMP converge with similar rates in the various regimes, namely with a sublinear rate for smooth convex functions and with a linear rate for

smooth strongly convex functions, however they have different advantages: GMP converges (much) faster in wall-clock time while OMP offers (much) sparser iterates. The interest in these algorithms stems from the fact that they work in the general setting of smooth convex functions in Hilbert spaces and that their convergence analyses do not require incoherence or restricted isometry properties (RIP, Candès and Tao [2005]) of the dictionary, which are quite strong assumptions from an optimization standpoint. For an in-depth discussion of the advantages of GMP and OMP over other methods, e.g., in Tropp [2004], Gribonval and Vandergheynst [2006], Davenport and Wakin [2010], Shalev-Shwartz et al. [2010], Temlyakov [2013, 2014, 2015], Tibshirani [2015], Yao and Kwok [2016], and Nguyen and Petrova [2017], we refer the interested reader to Locatello et al. [2017]. In a follow-up work, Locatello et al. [2018] presented an Accelerated Matching Pursuit algorithm.

We aim at unifying the best of GMP (speed) and OMP (sparsity) into a single algorithm by blending them strategically. However, while the overall idea is reasonably natural, we face considerable challenges as many important features of Frank-Wolfe methods do not apply anymore in the Matching Pursuit setting and cannot be as easily overcome as in Locatello et al. [2017], requiring a different analysis. For example, Frank-Wolfe (duality) gaps are not readily available but they are crucial in monitoring the blending, and further key components, such as the weak-separation oracle, require modifications.

**Contributions.** We propose a *Blended Matching Pursuit* algorithm (BMP), a fast and sparse first-order method for solving Problem (1). Our method unifies the best of GMP (speed) and OMP (sparsity) into one algorithm, which is of fundamental interest for practitioners. We establish a continuous range of convergence rates between $\mathcal{O}(1/\epsilon^p)$ and $\mathcal{O}(\ln 1/\epsilon)$, where $\epsilon > 0$ is the desired accuracy and $p > 0$ depends on the properties of the function. In particular, we derive linear rates of convergence for a large class of smooth convex but non-strongly convex functions. Lastly, we demonstrate the computational superiority of BMP over state-of-the-art methods, with BMP converging the fastest in wall-clock time while maintaining its iterates at close-to-optimal sparsity, and this without requiring sparsity-inducing constraints.

**Outline.** We introduce notions and notation in Section 2. We present the Blended Matching Pursuit algorithm in Section 3 with the convergence analyses in Section 3.1. Computational experiments are provided in Section 4. Additional experiments and the proofs can be found in the Appendix.

## 2 Preliminaries

We work in a separable real Hilbert space $(\mathcal{H}, \langle \cdot, \cdot \rangle)$ with induced norm $\| \cdot \|$. A set $\mathcal{D} \subset \mathcal{H}$ of normalized vectors is a *dictionary* if it is at most countable and $\mathrm{cl}(\mathrm{span}(\mathcal{D})) = \mathcal{H}$, and in this case its elements are referred to as *atoms*. For any set $\mathcal{S} \subseteq \mathcal{H}$, let $\mathcal{S}' := \mathcal{S} \cup -\mathcal{S}$ denote the *symmetrization of $\mathcal{S}$* and $D_{\mathcal{S}} := \sup_{u,v \in \mathcal{S}} \|u - v\|$ denote the *diameter of $\mathcal{S}$*. If $\mathcal{S}$ is closed and convex, let $\mathrm{proj}_{\mathcal{S}}$ denote the *orthogonal projection onto $\mathcal{S}$* and $\mathrm{dist}(\cdot, \mathcal{S}) := \|\mathrm{id} - \mathrm{proj}_{\mathcal{S}}\|$ denote the *distance to $\mathcal{S}$*. For Problem (1) to be feasible, we will assume $f$ to be *coercive*, i.e., $\lim_{\|x\| \to +\infty} f(x) = +\infty$. Since $f$ is convex, this is actually a mild assumption when $\arg\min_{\mathcal{H}} f \neq \varnothing$.

Let $f : \mathcal{H} \to \mathbb{R}$ be a Fréchet differentiable function. In the following, we use extended notions of smoothness and strong convexity by introducing *orders*, and we weaken and generalize the notion of strong convexity to that of *sharpness* (see, e.g., Roulet and d'Aspremont [2017] and Kerdreux et al. [2019] for recent work). We say that $f$ is:

(i) *smooth of order $\ell > 1$* if there exists $L > 0$ such that for all $x, y \in \mathcal{H}$,
$$f(y) - f(x) - \langle \nabla f(x), y - x \rangle \leqslant \frac{L}{\ell} \|y - x\|^{\ell},$$

(ii) *strongly convex of order $s > 1$* if there exists $S > 0$ such that for all $x, y \in \mathcal{H}$,
$$f(y) - f(x) - \langle \nabla f(x), y - x \rangle \geqslant \frac{S}{s} \|y - x\|^{s},$$

(iii) *sharp of order $\theta \in \,]0, 1[$ on $\mathcal{K}$* if $\mathcal{K} \subset \mathcal{H}$ is a bounded set, $\varnothing \neq \arg\min_{\mathcal{H}} f \subset \mathrm{int}(\mathcal{K})$, and there exists $C > 0$ such that for all $x \in \mathcal{K}$,
$$\mathrm{dist}\left(x, \arg\min_{\mathcal{H}} f\right) \leqslant C \left(f(x) - \min_{\mathcal{H}} f\right)^{\theta}.$$

If needed, we may specify the constants by introducing $f$ as *L-smooth*, *S-strongly convex*, or *C-sharp*. The following fact, whose result was already used in Nemirovskii and Nesterov [1985], provides a bound on the sharpness order of a smooth function. A proof is available in Appendix D.

**Fact 2.1.** *Let $f : \mathcal{H} \to \mathbb{R}$ be smooth of order $\ell > 1$, convex, and sharp of order $\theta \in ]0, 1[$ on $\mathcal{K}$. Then $\theta \in ]0, 1/\ell]$.*

## 2.1  On sharpness and strong convexity

Notice that if $f : \mathcal{H} \to \mathbb{R}$ is Fréchet differentiable and strongly convex of order $s > 1$, then $\operatorname{card}(\arg\min_{\mathcal{H}} f) = 1$. Let $\{x^*\} \coloneqq \arg\min_{\mathcal{H}} f$. It follows directly from $\nabla f(x^*) = 0$ that for any bounded set $\mathcal{K} \subset \mathcal{H}$ such that $x^* \in \operatorname{int}(\mathcal{K})$, $f$ is sharp of order $\theta = 1/s$ on $\mathcal{K}$. Thus, strong convexity implies sharpness. However, not every sharp function is strongly convex; moreover, the next example shows that not every sharp and convex function is strongly convex.

**Example 2.2 (Distance to a convex set).** *Let $\mathcal{C} \subset \mathcal{H}$ be a nonempty, closed, and bounded convex set, and $\mathcal{K} \subset \mathcal{H}$ be a bounded set such that $\mathcal{C} \subset \operatorname{int}(\mathcal{K})$. The function $f : x \in \mathcal{H} \mapsto \operatorname{dist}(x, \mathcal{C})^2 = \|x - \operatorname{proj}_{\mathcal{C}}(x)\|^2$ is convex, and it is sharp of order $\theta = 1/2$ on $\mathcal{K}$. Indeed, since $\arg\min_{\mathcal{H}} f = \mathcal{C}$ and $\min_{\mathcal{H}} f = 0$, we have for all $x \in \mathcal{K}$,*

$$\operatorname{dist}\left(x, \arg\min_{\mathcal{H}} f\right) = \|x - \operatorname{proj}_{\mathcal{C}}(x)\| = \left(f(x) - \min_{\mathcal{H}} f\right)^{1/2}.$$

*Now, suppose $\mathcal{C}$ contains more than one element. Then, $f$ has more than one minimizer. However, a function that is strongly convex of order $s > 1$ has no more than one minimizer. Therefore, $f$ cannot be strongly convex of order $s$, for all $s > 1$. Notice that $f$ is also a smooth function, of order $\ell = 2$.*

Hence, sharpness is a more general notion of strong convexity. It is a local condition around the optimal solutions while strong convexity is a global condition. In fact, building on the Łojasiewicz inequality of Łojasiewicz [1963], [Bolte et al., 2007, Equation (15)] showed that sharpness always holds in finite dimensional spaces for reasonably well-behaved convex functions; see Lemma 2.3. Polynomial convex functions, the $\ell_p$-norms, the Huber loss (see Appendix A.4), and the rectifier ReLU are simple examples of such functions.

**Lemma 2.3.** *Let $f : \mathbb{R}^n \to ]-\infty, +\infty]$ be a lower semicontinuous, convex, and subanalytic function with $\{x \in \mathbb{R}^n \mid 0 \in \partial f(x)\} \neq \varnothing$. Then for any bounded set $\mathcal{K} \subset \mathbb{R}^n$, there exists $\theta \in ]0, 1[$ and $C > 0$ such that for all $x \in \mathcal{K}$,*

$$\operatorname{dist}\left(x, \arg\min_{\mathbb{R}^n} f\right) \leqslant C \left(f(x) - \min_{\mathbb{R}^n} f\right)^{\theta}.$$

Strong convexity is a standard requirement to prove linear convergence rates on smooth convex objectives but, regrettably, this considerably restricts the set of candidate functions. For our Blended Matching Pursuit algorithm, we will only require sharpness to establish linear convergence rates, thus including a larger class of functions.

## 2.2  Matching Pursuit algorithms

For $y \in \mathcal{H}$ and $f : x \in \mathcal{H} \mapsto \|y - x\|^2/2$, Problem (1) falls in the area of sparse recovery and is often solved with the Matching Pursuit algorithm [Mallat and Zhang, 1993]. The algorithm recovers a sparse representation of the signal $y$ from the dictionary $\mathcal{D}$ by sequentially *pursuing* the best *matching* atom. At each iteration, it searches for the atom $v_t \in \mathcal{D}$ most correlated with the residual $y - x_t$, i.e., $v_t \coloneqq \arg\max_{v \in \mathcal{D}} |\langle y - x_t, v \rangle|$, and adds it to the linear decomposition of the current iterate $x_t$ to form the new iterate $x_{t+1}$, keeping track of the *active set* $\mathcal{S}_{t+1} = \mathcal{S}_t \cup \{v_t\}$. However, this does not prevent the algorithm from selecting atoms that have already been added in earlier iterations or that are redundant, hence affecting sparsity. The Orthogonal Matching Pursuit variant [Pati et al., 1993, Davis et al., 1994] overcomes this by computing the new iterate as the projection of the signal $y$ onto $\mathcal{S}_t \cup \{v_t\}$; see Chen et al. [1989] and Tropp [2004] for analyses and Zhang [2009] for an extension to the stochastic case. Thus, $y - x_{t+1}$ becomes orthogonal to the active set.

In order to solve Problem (1) for any smooth convex objective, Locatello et al. [2017] proposed the Generalized Matching Pursuit (GMP) and Generalized Orthogonal Matching Pursuit (GOMP) algorithms (Algorithm 1); slightly abusing notation we will refer to the latter simply as Orthogonal

Matching Pursuit (OMP). The atom selection subroutine is implemented with a Frank-Wolfe linear minimization oracle $\arg\min_{v\in\mathcal{D}'}\langle\nabla f(x_t),v\rangle$ (Line 3). The solution $v_t$ to this oracle is guaranteed to be a descent direction as it satisfies $\langle\nabla f(x_t),v_t\rangle\leqslant 0$ by symmetry of $\mathcal{D}'$, and $\langle\nabla f(x_t),v_t\rangle=0$ if and only if $x_t\in\arg\min_{\mathcal{H}}f$. Notice that for $y\in\mathcal{H}$ and $f:x\in\mathcal{H}\mapsto\|y-x\|^2/2$, the GMP and OMP variants of Algorithm 1 recover the original Matching Pursuit and Orthogonal Matching Pursuit algorithms respectively. In particular, up to a sign which does not affect the sequence of iterates, $\arg\max_{v\in\mathcal{D}}|\langle y-x_t,v\rangle|\Leftrightarrow\arg\min_{v\in\mathcal{D}'}\langle\nabla f(x_t),v\rangle$. In practice, the main difference in the case of general smooth convex functions is that the OMP variant (Line 6) is much more expensive, as a closed-form solution to this projection step is not available anymore. Hence, Line 6 is typically a sequence of projected gradient steps and OMP is significantly slower than GMP to converge.

---

**Algorithm 1** Generalized/Orthogonal Matching Pursuit (GMP/OMP)

---

**Input:** Start atom $x_0\in\mathcal{D}$, number of iterations $T\in\mathbb{N}^*$.
**Output:** Iterates $x_1,\ldots,x_T\in\mathrm{span}(\mathcal{D})$.

1: $\mathcal{S}_0\leftarrow\{x_0\}$
2: **for** $t=0$ **to** $T-1$ **do**
3: $\quad v_t\leftarrow\arg\min_{v\in\mathcal{D}'}\langle\nabla f(x_t),v\rangle$
4: $\quad\mathcal{S}_{t+1}\leftarrow\mathcal{S}_t\cup\{v_t\}$
5: $\quad$*GMP variant*: $x_{t+1}\leftarrow\arg\min_{x_t+\mathbb{R}v_t}f$
6: $\quad$*OMP variant*: $x_{t+1}\leftarrow\arg\min_{\mathrm{span}(\mathcal{S}_{t+1})}f$
7: **end for**

---

### 2.3 Weak-separation oracle

We present in Oracle 2 the weak-separation oracle, a *modified* version of the one first introduced in Braun et al. [2017] and used in, e.g., Lan et al. [2017], Braun et al. [2019]. Note that the modification asks for an unconstrained improvement, whereas the original weak-separation oracle required an improvement relative to a reference point. As such, our variant here is even simpler than the original weak-separation oracle. The oracle is called in Line 11 by the Blended Matching Pursuit algorithm.

---

**Oracle 2** Weak-separation $\mathrm{LPsep}_{\mathcal{D}}(c,\phi,\kappa)$

---

**Input:** Linear objective $c\in\mathcal{H}$, objective value $\phi\leqslant 0$, accuracy $\kappa\geqslant 1$.
**Output:** Either atom $v\in\mathcal{D}$ such that $\langle c,v\rangle\leqslant\phi/\kappa$ (positive call), or **false** ensuring $\langle c,z\rangle\geqslant\phi$ for all $z\in\mathrm{conv}(\mathcal{D})$ (negative call).

---

The weak-separation oracle determines whether there exists an atom $v\in\mathcal{D}$ such that $\langle c,v\rangle\leqslant\phi/\kappa$, and thereby relaxes the Frank-Wolfe linear minimization oracle. If not, then this implies that $\mathrm{conv}(\mathcal{D})$ can be *separated* from the ambient space by $c$ and $\phi$ with the linear inequality $\langle c,z\rangle\geqslant\phi$ for all $z\in\mathrm{conv}(\mathcal{D})$. In practice, the oracle can be efficiently implemented using *caching*, i.e., first testing atoms that were already returned during previous calls as they may satisfy the condition here again. In this case, caching also preserves sparsity. If no active atom satisfies the condition, the oracle can be solved, e.g., by means of a call to a linear optimization oracle; see Braun et al. [2017] for an in-depth discussion. Lastly, we would like to briefly note that the parameter $\kappa$ can be used to further promote positive calls over negative calls, by weakening the improvement requirement and therefore speeding up the oracle. Indeed, only negative calls need a full scan of the dictionary.

## 3 The Blended Matching Pursuit algorithm

We now present our Blended Matching Pursuit algorithm (BMP) in Algorithm 3. Note that although we blend steps, we maintain the explicit decomposition of the iterates $x_t=\sum_{j=1}^{n_t}\lambda_{t,i_j}a_{i_j}$ as linear combinations of the atoms.

**Remark 3.1 (Algorithm design).** *BMP actually does not require the atoms to have exactly the same norm and only needs the dictionary to be bounded, whether it be for ensuring the convergence rates or for computations; one could further take advantage of this to add weights to certain atoms. Line 6*

---

**Algorithm 3** Blended Matching Pursuit (BMP)

---

**Input:** Start atom $x_0 \in \mathcal{D}$, parameters $\eta > 0$, $\kappa \geqslant 1$, and $\tau > 1$, number of iterations $T \in \mathbb{N}^*$.
**Output:** Iterates $x_1, \ldots, x_T \in \operatorname{span}(\mathcal{D})$.

1:   $\mathcal{S}_0 \leftarrow \{x_0\}$
2:   $\phi_0 \leftarrow \min\limits_{v \in \mathcal{D}'} \langle \nabla f(x_0), v \rangle / \tau$
3:   **for** $t = 0$ **to** $T - 1$ **do**
4:      $v_t^{\text{FW-}\mathcal{S}} \leftarrow \arg\min\limits_{v \in \mathcal{S}_t'} \langle \nabla f(x_t), v \rangle$
5:      **if** $\left\langle \nabla f(x_t), v_t^{\text{FW-}\mathcal{S}} \right\rangle \leqslant \phi_t / \eta$ **then**
6:         $\widetilde{\nabla} f(x_t) \leftarrow \operatorname{proj}_{\operatorname{span}(\mathcal{S}_t)}(\nabla f(x_t))$
7:         $x_{t+1} \leftarrow \arg\min\limits_{x_t + \mathbb{R}\widetilde{\nabla} f(x_t)} f$                           {constrained step}
8:         $\mathcal{S}_{t+1} \leftarrow \mathcal{S}_t$
9:         $\phi_{t+1} \leftarrow \phi_t$
10:     **else**
11:        $v_t \leftarrow \operatorname{LPsep}_{\mathcal{D}'}(\nabla f(x_t), \phi_t, \kappa)$
12:        **if** $v_t = $ **false then**
13:           $x_{t+1} \leftarrow x_t$                                       {dual step}
14:           $\mathcal{S}_{t+1} \leftarrow \mathcal{S}_t$
15:           $\phi_{t+1} \leftarrow \phi_t / \tau$
16:        **else**
17:           $x_{t+1} \leftarrow \arg\min\limits_{x_t + \mathbb{R}v_t} f$                       {full step}
18:           $\mathcal{S}_{t+1} \leftarrow \mathcal{S}_t \cup \{v_t\}$
19:           $\phi_{t+1} \leftarrow \phi_t$
20:        **end if**
21:     **end if**
22:     *Optional:* Correct $\mathcal{S}_{t+1}$
23: **end for**

---

*is simply taking the component of $\nabla f(x_t)$ parallel to $\operatorname{span}(\mathcal{S}_t)$, which can be achieved by basic linear algebra and costs $\mathcal{O}(n \operatorname{card}(\mathcal{S}_t)^2)$ when $\mathcal{H} = \mathbb{R}^n$. The line searches Lines 7 and 17 can be replaced with explicit step sizes using the smoothness of $f$ (see Fact B.2 in the Appendix). The purpose of (the optional) Line 22 is to reoptimize the active set $\mathcal{S}_{t+1}$, e.g., by reducing it to a subset that forms a basis for its linear span. One could also obtain further sparsity by removing atoms whose coefficient in the decomposition of the iterate is smaller than some threshold $\delta > 0$.*

**Blending.** BMP aims at unifying the best of GMP and OMP. As seen in Section 2.2, an OMP iteration is a sequence of projected gradient (PG) steps. The idea is that the sequence of PG steps constituting an OMP iteration is actually overkill: there is a sweet spot where further optimizing over $\operatorname{span}(\mathcal{S}_t)$ is less effective than adding a new atom and taking a GMP step into a (possibly) new space. However, PG steps have the benefit of preserving sparsity, since no new atom is added. Furthermore, GMP steps require an expensive scan of the dictionary to output the descent direction $v_t^{\text{FW}} \leftarrow \arg\min_{v \in \mathcal{D}'} \langle \nabla f(x_t), v \rangle$. To remedy this, BMP blends *constrained steps* (PG steps, Line 7) with *full steps* (lazified GMP steps, Line 17) by promoting constrained steps as long as the progress in function value is *comparable* to that of a GMP step, else by taking a full step in an approximate direction $v_t$ (with cheap computation via Oracle 2) such that the progress is *comparable* to that of a GMP step. Therefore, to monitor this blending of steps, we wish to compare $\langle \nabla f(x_t), v_t^{\text{FW-}\mathcal{S}} \rangle$ and $\langle \nabla f(x_t), v_t \rangle$ to $\langle \nabla f(x_t), v_t^{\text{FW}} \rangle$, which quantities measure the progress in function value offered by a constrained step, a full step, and a GMP step respectively (see proofs in the Appendix).

**Dual gap estimates.** The aforementioned comparisons however cannot be made directly as the quantity $\langle \nabla f(x_t), v_t^{\text{FW}} \rangle$ is (deliberately) not computed; computing it requires an expensive complete scan of the dictionary. Instead, we use an estimation of this quantity, by introducing the *dual gap estimate* $|\phi_t|$. This designation comes from the fact that $-\langle \nabla f(x_t), v_t^{\text{FW}} \rangle$ is our equivalent of the *duality gap* from the constrained setting (see, e.g., Jaggi [2013]), and this will guide how we build our estimation. Indeed, since $\mathcal{D}'$ is symmetric and assuming $0 \in \operatorname{int}(\operatorname{conv}(\mathcal{D}'))$, there exists (an unknown)

$\rho > 0$ such that $\{x_0, \ldots, x_T\} \cup \arg\min_{\mathcal{H}} f \subset \rho \operatorname{conv}(\mathcal{D}')$. Then for all $x^* \in \arg\min_{\mathcal{H}} f$,

$$\epsilon_t := f(x_t) - f(x^*) \leqslant \langle \nabla f(x_t), x_t - x^* \rangle$$

$$\leqslant \max_{u,v \in \rho\operatorname{conv}(\mathcal{D}')} \langle \nabla f(x_t), u - v \rangle = -2\rho \langle \nabla f(x_t), v_t^{\mathrm{FW}} \rangle, \qquad (2)$$

which is our desired inequality. We set $\phi_0 \leftarrow \langle \nabla f(x_0), v_0^{\mathrm{FW}} \rangle / \tau$ (Line 2) so $\epsilon_0 \leqslant 2\tau\rho|\phi_0|$ by (2). The criterion in Line 5 compares $\langle \nabla f(x_t), v_t^{\mathrm{FW}\text{-}\mathcal{S}} \rangle$ to $\phi_t$. If this quantity is below the threshold $\phi_t$, then a constrained step is not taken and the weak-separation oracle (Line 11, Oracle 2) is called to search for an atom $v_t$ satisfying $\langle \nabla f(x_t), v_t \rangle \leqslant \phi_t$. If the oracle cannot find such an atom, then a full step is not taken and it returns a *negative call* with the certificate $\langle \nabla f(x_t), v_t^{\mathrm{FW}} \rangle > \phi_t$. In this case, BMP has detected an improved dual gap estimate and takes a *dual step* (Line 13): by (2), this implies that $\epsilon_t \leqslant 2\rho|\phi_t|$ so with $\phi_{t+1} \leftarrow \phi_t/\tau$ and $x_{t+1} \leftarrow x_t$, we recover $\epsilon_{t+1} \leqslant 2\tau\rho|\phi_{t+1}|$. Furthermore, observe that this update is a geometric rescaling which ensures that BMP requires only $N_{\mathrm{dual}} = \mathcal{O}(\ln 1/\epsilon)$ dual steps (see proofs). Thus, the total number of negative calls, i.e., the number of iterations requiring a complete scan of the dictionary, is only $\mathcal{O}(\ln 1/\epsilon)$. Therefore, for this and for the blending of steps, the dual gap estimates $|\phi_t|$ are the key to the speed-up realized by BMP.

**Parameters.** BMP involves three (hyper-)parameters $\eta > 0$, $\kappa \geqslant 1$, and $\tau > 1$ to be set before running the algorithm. The parameter $\eta$ needs to be tuned carefully, as its value affects the criterion in Line 5 to promote either speed of convergence (e.g., $\eta \sim 0.1$, promoting full steps) or sparsity of the iterates (e.g., $\eta \sim 1000$, promoting constrained steps). In our experiments (see Section 4 and the Appendix), we found that setting $\eta \sim 5$ leads to close to both maximal speed of convergence and sparsity of the iterates, with the default choices $\kappa = \tau = 2$. In this setting, BMP converges (much) faster than GMP and has iterates with sparsity very comparable to that of OMP, and therefore it is possible to enjoy both properties of speed and sparsity simultaneously. Note that the value of $\kappa$ also impacts the range of values of $\eta$ to which BMP is sensitive, since the criterion (Line 5) tests $\min_{v \in \mathcal{S}'_t} \langle \nabla f(x_t), v \rangle \leqslant \phi_t/\eta$ while the weak-separation oracle asks for $v \in \mathcal{D}'$ such that $\langle \nabla f(x_t), v \rangle \leqslant \phi_t/\kappa$. In specific experiments, parameter tuning might further improve performance.

## 3.1 Convergence analyses

We start with the simpler case of smooth convex functions of order $\ell > 1$ (Theorem 3.2). Our main result is Theorem 3.3, which subsumes the case of strongly convex functions. To establish the convergence rates of GMP and OMP, Locatello et al. [2017] assume knowledge of an upper bound on $\sup\{\|x^*\|_{\mathcal{D}'}, \|x_0\|_{\mathcal{D}'}, \ldots, \|x_T\|_{\mathcal{D}'}\}$ where $\|\cdot\|_{\mathcal{D}'} : x \in \mathcal{H} \mapsto \inf\{\rho > 0 \mid x \in \rho\operatorname{conv}(\mathcal{D}')\}$ is the *atomic norm*. In Locatello et al. [2018], this is resolved by working with the atomic norm $\|\cdot\|_{\mathcal{D}'}$ instead of the Hilbert space induced norm $\|\cdot\|$ to, e.g., define smoothness and strong convexity of $f$ and derive the proofs, but $\|\cdot\|_{\mathcal{D}'}$ itself can be difficult to derive in many applications. In contrast, we need neither the finiteness assumption nor to change the norm, however we assume $f$ to be coercive to ensure feasibility of Problem (1), a reasonably mild assumption.

**Theorem 3.2 (Smooth convex case).** *Let $\mathcal{D} \subset \mathcal{H}$ be a dictionary such that $0 \in \operatorname{int}(\operatorname{conv}(\mathcal{D}'))$ and let $f : \mathcal{H} \to \mathbb{R}$ be smooth of order $\ell > 1$, convex, and coercive. Then the Blended Matching Pursuit algorithm (Algorithm 3) ensures that $f(x_t) - \min_{\mathcal{H}} f \leqslant \epsilon$ for all $t \geqslant T$ where $T = \mathcal{O}\big((L/\epsilon)^{1/(\ell-1)}\big)$.*

We now present our main result in its full generality. We provide the general convergence rates of BMP (Algorithm 3) in Theorem 3.3. Recall that sharpness is implied by strong convexity and that it is a very mild assumption in finite dimensional spaces as it is satisfied by all *well-behaved* convex functions (Lemma 2.3).

**Theorem 3.3 (Smooth convex sharp case).** *Let $\mathcal{D} \subset \mathcal{H}$ be a dictionary such that $0 \in \operatorname{int}(\operatorname{conv}(\mathcal{D}'))$ and let $f : \mathcal{H} \to \mathbb{R}$ be $L$-smooth of order $\ell > 1$, convex, coercive, and $C$-sharp of order $\theta \in \,]0, 1/\ell]$ on $\mathcal{K}$. Then the Blended Matching Pursuit algorithm (Algorithm 3) ensures that $f(x_t) - \min_{\mathcal{H}} f \leqslant \epsilon$ for all $t \geqslant T$ where*

$$T = \begin{cases} \mathcal{O}\left( C^{1/(1-\theta)} L^{1/(\ell-1)} \ln\left( \dfrac{C|\phi_0|}{\epsilon^{1-\theta}} \right) \right) & \text{if } \ell\theta = 1 \\[1.5em] \mathcal{O}\left( \left( \dfrac{C^\ell L}{\epsilon^{1-\ell\theta}} \right)^{1/(\ell-1)} \right) & \text{if } \ell\theta < 1. \end{cases}$$

*Moreover, $\operatorname{dist}(x_t, \arg\min_{\mathcal{H}} f) \to 0$ as $t \to +\infty$ at same rate.*

If $f$ is not strongly convex then Locatello et al. [2017] only guarantee a sublinear convergence rate $\mathcal{O}(1/\epsilon)$ for GMP and OMP, while Theorem 3.3 can still guarantee higher convergence rates, up to linear convergence $\mathcal{O}(\ln 1/\epsilon)$ if $\ell\theta = 1$, using sharpness. Note that in the popular case of smooth strongly convex functions of orders $\ell = 2$ and $s = 2$, Theorem 3.3 guarantees a linear convergence rate as these functions are sharp of order $\theta = 1/2$ (with constant $C = \sqrt{2/S}$) and thus satisfy $\ell\theta = 1$. For completeness, we also study this special case in Appendix C, with a simpler proof. In conclusion, Theorem 3.3 extends linear convergence rates to a large class of non-strongly convex functions solving Problem (1).

**Remark 3.4 (Optimality of the convergence rates).** *Let $n \leqslant +\infty$ be the dimension of $\mathcal{H}$. Nemirovskii and Nesterov [1985] provided unimprovable rates when solving Problem (1) in different cases. These optimal rates are reported in Table 1, where we compare them to those of BMP proved in this paper (Theorems 3.2 and 3.3). The third column gives the lower bounds on complexity stated in Nemirovskii and Nesterov [1985, Equations (1.20), (1.21'), and (1.21)]. Note that our rates are dimension independent and hold globally across iterations. It remains an open question to determine whether the gap in the exponent can be closed by accelerating BMP.*

Table 1: Comparison of the rates of BMP vs. the lower bounds on complexity.

| Properties of $f$ | BMP rate | Lower bound on complexity |
|---|---|---|
| Smooth convex | $T(\epsilon) = \mathcal{O}\left(\dfrac{1}{\epsilon^{1/(\ell-1)}}\right)$ | $T(\epsilon) = \Omega\left(\min\left\{n, \dfrac{1}{\epsilon^{1/(1.5\ell-1)}}\right\}\right)$ |
| Smooth convex sharp with $\ell = 2, \theta = 1/2$ | $T(\epsilon) = \mathcal{O}\left(\ln\left(\dfrac{1}{\epsilon}\right)\right)$ | $T(\epsilon) = \Omega\left(\min\left\{n, \ln\left(\dfrac{1}{\epsilon}\right)\right\}\right)$ |
| Smooth convex sharp with $\ell\theta < 1$ | $T(\epsilon) = \mathcal{O}\left(\dfrac{1}{\epsilon^{(1-\ell\theta)/(\ell-1)}}\right)$ | $T(\epsilon) = \Omega\left(\min\left\{n, \dfrac{1}{\epsilon^{(1-\ell\theta)/(1.5\ell-1)}}\right\}\right)$ |

## 4 Computational experiments

We implemented BMP in Python 3 along with GMP and OMP [Locatello et al., 2017], the Accelerated Matching Pursuit algorithm (accMP) [Locatello et al., 2018], and the Blended Conditional Gradients (BCG) [Braun et al., 2019] and Conditional Gradient with Enhancement and Truncation (CoGEnT) [Rao et al., 2015] algorithms for completeness. All algorithms share the same code framework to ensure fair comparison. No enhancement beyond basic coding was performed. We ran the experiments on a laptop under Linux Ubuntu 18.04 with Intel Core i7 3.5GHz CPU and 8GB RAM. The random data are drawn from Gaussian distributions. For GMP, OMP, BCG, and CoGEnT, we represented the dual gaps by $-\min_{v\in\mathcal{D}'}\langle\nabla f(x_t), v\rangle$, yielding a zig-zag plot dissimilar to the stair-like plot of the dual gap estimates $|\phi_t|$ of BMP. The Appendix contains additional experiments.

### 4.1 Comparison of BMP vs. GMP, OMP, BCG, and CoGEnT

Let $\mathcal{H}$ be the Euclidean space $(\mathbb{R}^n, \langle\cdot,\cdot\rangle)$ and $\mathcal{D}$ be the set of signed canonical vectors $\{\pm e_1, \ldots, \pm e_n\}$. Suppose we want to learn the (sparse) source $x^*$ from observed data $y := Ax^* + w$, where $A \in \mathbb{R}^{m\times n}$ and where $w \sim \mathcal{N}(0, \sigma^2 I_m)$ is the noise in the observed $y$. The general and most intuitive formulation of the problem is $\min_{x\in\mathbb{R}^n} \|y - Ax\|_2^2$ s.t. $\|x\|_0 \leqslant \|x^*\|_0 =: s$ but the $\ell_0$-pseudo norm constraint is nonconvex and makes the problem NP-hard and therefore intractable in many situations [Natarajan, 1995]. To remedy this, one can handle the sparsity constraint in various ways, either by completely removing it and relying on an algorithm inherently promoting sparsity (as done in BMP, GMP, and OMP), or through a convex relaxation of the constraint, often via the $\ell_1$-norm, and then solving the new constrained convex problem $\min_{x\in\mathbb{R}^n} \|y - Ax\|_2^2$ s.t. $\|x\|_1 \leqslant \|x^*\|_1$ (as done in BCG and CoGEnT). We ran a comparison of these methods, where we favorably provided the constraint $\|x\|_1 \leqslant \|x^*\|_1$ for BCG and CoGEnT although $x^*$ is unknown. We set $m = 500$, $n = 2000$, $s = 100$, and $\sigma = 0.05$. In BMP, we set $\kappa = \tau = 2$ and we chose $\eta = 5$; see Appendix A.1 for an in-depth sensitivity analysis of BMP with respect to $\eta$. We did not perform any additional correction of the active sets (Line 22). Note that [Rao et al., 2015, Table III] demonstrated the superiority of

CoGEnT over CoSaMP [Needell and Tropp, 2009], Subspace Pursuit [Dai and Milenkovic, 2009], and Gradient Descent with Sparsification [Garg and Khandekar, 2009] on an equivalent experiment and we therefore do not compare to those methods.

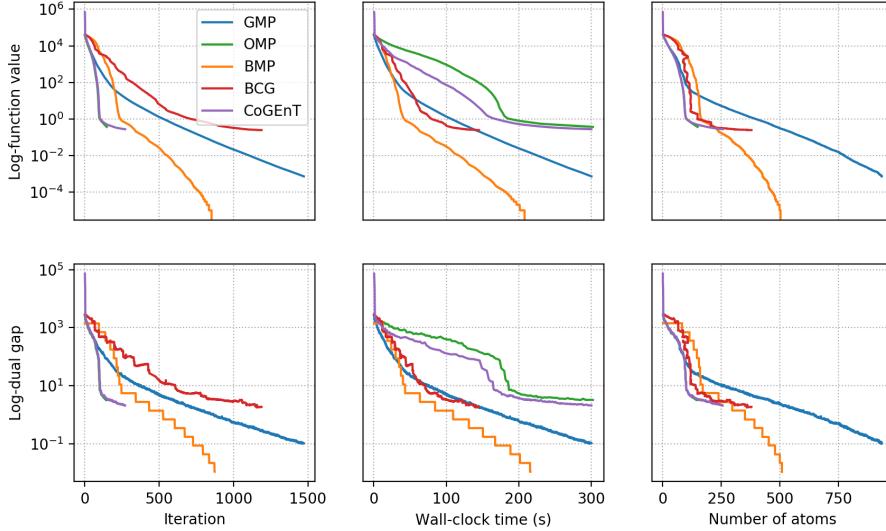

Figure 1: Comparison of BMP vs. GMP, OMP, BCG, and CoGEnT, with $\eta = 5$.

Figure 1 shows that BMP is the fastest algorithm in wall-clock time and has close-to-optimal sparsity. It is important to stress that, unlike BCG and CoGEnT, BMP achieves this while having no explicit sparsity-promoting constraint, regularization, nor information on $x^*$. Thus, when $\|x^*\|_1$ is not provided, which is the case in most applications, BCG and CoGEnT would require a hyper-parameter tuning of the sparsity-inducing constraint (or, equivalently, the Lagrangian penalty parameters), such as the radius of the $\ell_1$-ball [Tibshirani, 1996], as used here, or the trace-norm-ball [Fazel et al., 2001]. OMP and CoGEnT converge faster per-iteration, as expected, given that they solve a reoptimization problem at each iteration, however this is very costly and the disadvantage becomes evident in wall-clock time performance. Note that another "obvious" choice for an algorithm would be projected gradient descent, however the provided sparsity is far from sufficient (see Appendix A.2).

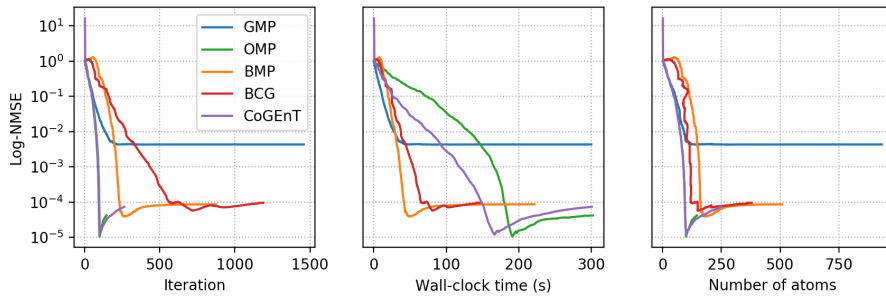

Figure 2: Comparison in NMSE of BMP vs. GMP, OMP, BCG, and CoGEnT, with $\eta = 5$.

In Figure 2, we compare the Normalized Mean Squared Error (NMSE) of the different methods. The NMSE at iterate $x_t$ is defined as $\|x_t - x^*\|_2^2 / \|x^*\|_2^2$. The plots show a rebound occurring once the NMSE reaches $\sim 10^{-4}$, which is due to the algorithms overfitting to the noisy measurements $y$. A post-processing step can mitigate the rebound via early stopping or by removing atoms whose coefficient in the decomposition of the iterate are smaller than some threshold $\delta > 0$. We used early stopping on a validation set and present the test error $\|y_{\text{test}} - A_{\text{test}} x_T\|_2^2 / m_{\text{test}}$ on a test set in Table 2, where $x_T$ is the solution iterate for each algorithm. For completeness, we also reported the results for the Gradient Hard Thresholding Pursuit (GraHTP) and Fast Gradient Hard Thresolding Pursuit (Fast GraHTP) algorithms [Yuan et al., 2018], for which we favorably set $k = \|x^*\|_0$. As expected, GMP

performs the worst on the test set because its NMSE does not achieve sufficient convergence (see Figure 2), highlighting the importance of a clean, i.e., sparse, decomposition into the dictionary $\mathcal{D}$.

Table 2: Test error achieved using early stopping on a validation set.

| Algorithm | GMP | OMP | BMP | BCG | CoGEnT | GraHTP | Fast GraHTP |
|---|---|---|---|---|---|---|---|
| Test error | 0.1917 | 0.0036 | 0.0037 | 0.0068 | 0.0043 | 0.0036 | 0.0037 |

The Appendix contains additional experiments on different objective functions: an arbitrarily chosen norm (Appendix A.3), the Huber loss (Appendix A.4), the distance to a convex set (Appendix A.5), and a logistic regression loss (Appendix A.6). The conclusions are identical.

## 4.2   Comparison of BMP vs. accMP

Locatello et al. [2018] recently provided an Accelerated Matching Pursuit algorithm (accMP) for solving Problem (1). We implemented the same code as theirs, using the exact same parametrization. The code framework matches the one we used for BMP. We ran BMP on their toy data example and compared the results against accMP (which they labeled *accelerated steepest* in their plot); notice that we recovered their (per-iteration) plot exactly. The experiment is to minimize $f : x \in \mathbb{R}^{100} \mapsto \|x - b\|_2^2/2$ over the linear span of $\mathcal{D}$, where $\mathcal{D}$ is dictionary of 200 randomly chosen atoms in $\mathbb{R}^{100}$ and $b \in \mathbb{R}^{100}$ is also randomly chosen. The parameters, kindly provided by the authors of Locatello et al. [2018], for accMP were $L = 1000$ and $\nu = 1$. We report the results in Figure 3.

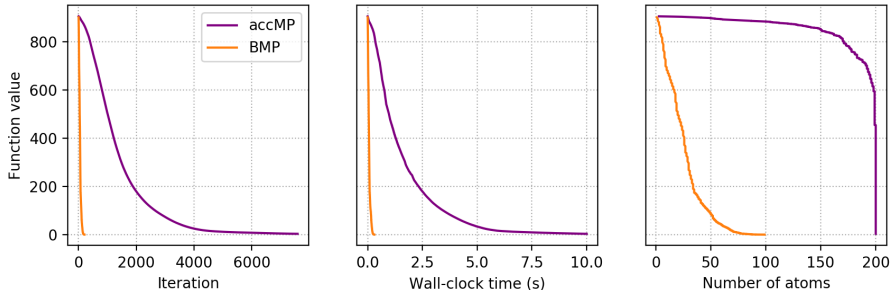

Figure 3: Comparison of BMP vs. accMP, with $\eta = 3$.

We see that BMP outperforms accMP in both speed of convergence and sparsity of the iterates. In fact, in terms of sparsity, accMP needs to use all available atoms to converge while BMP needs only half as much. Furthermore, accMP needs 75% of all available atoms to start converging significantly while BMP starts to converge instantaneously. We suspect that this is due to the following: accMP accelerates coordinate descent-like directions, which might be relatively bad approximations of the actual descent direction $-\nabla f(x_t)$, whereas BMP is working directly with (the projection of) $-\nabla f(x_t)$, achieving much more progress and offsetting the effect of acceleration.

## 5   Final remarks

We presented a Blended Matching Pursuit algorithm (BMP) which enjoys both properties of fast rate of convergence and sparsity of the iterates. More specifically, we derived linear convergence rates for a large class of non-strongly convex functions solving Problem (1), and we showed that our blending approach outperforms the state-of-the-art methods in speed of convergence while achieving close-to-optimal sparsity, and this without requiring sparsity-inducing constraints nor regularization. Although BMP already outperforms the Accelerated Matching Pursuit algorithm [Locatello et al., 2018] in our experiments, we believe it is also amenable to acceleration.

**Acknowledgments**

Research reported in this paper was partially supported by NSF CAREER award CMMI-1452463.

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
