[Supplementary Material]

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

# A   Additional computational experiments

We provide the sensitivity analysis to the experiment in Figure 1 in Appendix A.1, and the comparison to the projected gradient method in Appendix A.2. We then conduct additional experiments on a variety of objective functions: an arbitrarily chosen norm (Appendix A.3), the Huber loss (Appendix A.4), the distance to a convex set (Appendix A.5), and a logistic regression loss (Appendix A.6).

## A.1   Sensitivity of BMP to the parameter $\eta$

Here we report the sensitivity analysis of BMP for the data in Section 4.1. We ran BMP (Algorithm 3) for values of $\eta$ in $\{100, 10, 5, 2, 1\}$. We set $\kappa = 2$ and $\tau = 2$ and did not activate the correction of atoms (Line 22). We report the results in Figure 4.

Figure 4: Sensitivity of BMP to the parameter $\eta$.

We see that $\eta = 5$ is at the sweet spot between speed of convergence and sparsity of the iterates. Higher values of $\eta$ have similar levels of sparsity but they perform worse for speed of convergence. Lower values of $\eta$ perform much worse in sparsity and are not better in speed; $\eta = 2$ offsets $\eta = 5$ after 100 seconds but the function value is already $10^{-2}$ at that point. Therefore, by setting $\eta = 5$ in this example we achieve both speed of convergence and sparsity of the iterates. Similar insights are obtained in the other experiments, so that $\eta \sim 5$ seems to be a good initial choice.

For completeness, we present in Figure 5 the sensitivity of BMP to $\eta$ in NMSE.

Figure 5: Sensitivity of BMP to the parameter $\eta$ in NMSE.

## A.2 Comparison with PGD

Projected gradient descent (PGD) is a natural candidate for the experiment in Section 4.1. However, it does not ensure sufficient sparsity of the iterates. We depict three configurations of PGD in Figure 6, each named "PGD:$\alpha$" where PGD is ran with the constraint $\|x\|_1 \leqslant \alpha \|x^*\|_1$ and $\alpha \in \{1/2, 1, 2\}$. The implementation of PGD is in line with our general code framework and we used the method of Condat [2016] for projections onto the $\ell_1$-ball. The number of atoms collected by the iterates in PGD are reported as the number of nonzero coordinates. Note that in BMP, GMP, and OMP we do not check if a selected atom $v_t$ already satisfies $-v_t \in \mathcal{S}_t$ before adding it to $\mathcal{S}_t$, which is disadvantageous to these algorithms when evaluating their sparsity performance.

Figure 6: Comparison of PGD vs. the MP algorithms.

As expected, the constraint $\|x\|_1 \leqslant \|x^*\|_1$ provides the best results for PGD; the constraint $\|x\|_1 \leqslant 2\|x^*\|_1$ is too loose and basically produces no sparsity in the iterates (recall that the ambient space is $\mathbb{R}^{2000}$). In the configuration $\|x\|_1 \leqslant \|x^*\|_1$, PGD does not converge faster than OMP and produces significantly worse sparsity than OMP and BMP.

## A.3 Regression with arbitrarily chosen norm

We set $m = 250$, $n = 1000$, $s = 50$, and $\sigma = 0.05$ and generated the data as in Section 4.1. We ran a comparison with the arbitrarily chosen $f : x \in \mathbb{R}^n \mapsto \|Ax - b\|_3^5$. We plot the results in Figure 7.

Figure 7: Comparison of BMP vs. GMP and OMP on $f : x \in \mathbb{R}^n \mapsto \|Ax - b\|_3^5$, with $\eta = 5$.

We see that BMP offers very close-to-optimal levels of sparsity while being faster than the other algorithms in wall-clock time. We provide a sensitivity analysis of BMP to the parameter $\eta$ in Figure 8. We see that $\eta \sim 5$ is an appropriate choice combining the best of speed and sparsity.

Figure 8: Sensitivity of BMP to the parameter $\eta$.

For completeness, we present in Figure 9 the sensitivity of BMP to $\eta$ in NMSE.

Figure 9: Sensitivity of BMP to the parameter $\eta$ in NMSE.

## A.4 Huber loss

The Huber loss [Huber, 1964] is a smooth combination of the squared and absolute losses. The absolute loss is robust to outliers in the dataset, however its gradient is piecewise constant and not defined at the origin. This leads to instability of the solutions. The Huber loss overcomes this by behaving like the squared loss around the origin:

$$h_\delta : t \in \mathbb{R} \mapsto \begin{cases} t^2/2 & \text{if } |t| \leqslant \delta \\ \delta(|t| - \delta/2) & \text{else} \end{cases}$$

where $\delta > 0$ defines this region around the origin. We set $m = 250$, $n = 1000$, $s = 50$, and $\sigma = 0.05$ and generated the data as in Section 4.1. In this experiment we aim at minimizing the smooth convex function

$$f : x \in \mathbb{R}^n \mapsto \sum_{i=1}^m h_{10}(a_i^\top x - y_i)$$

where $a_1^\top, \ldots, a_m^\top \in \mathbb{R}^{1 \times n}$ are the rows of $A$. We plot the results in Figure 10.

Figure 10: Comparison of BMP vs. GMP and OMP on $f : x \in \mathbb{R}^n \mapsto \sum_{i=1}^m h_{10}(a_i^\top x - y_i)$, with $\eta = 5$.

Again, BMP has very close-to-optimal levels of sparsity while being the fastest algorithm to converge. We provide a sensitivity analysis of BMP to the parameter $\eta$ in Figure 11. We see that $\eta \sim 5$ is an appropriate choice combining the best of speed and sparsity.

Figure 11: Sensitivity of BMP to the parameter $\eta$.

For completeness, we present in Figure 12 the sensitivity of BMP to $\eta$ in NMSE.

Figure 12: Sensitivity of BMP to the parameter $\eta$ in NMSE.

## A.5 Distance to a convex set

Here we compared BMP vs. GMP and OMP on an arbitrarily chosen problem. We used

$$f : x \in \mathbb{R}^{500} \mapsto d_{\overline{\mathcal{B}}(0,1)}(Ax - b)^2 = \|(Ax - b) - \text{proj}_{\overline{\mathcal{B}}(0,1)}(Ax - b)\|_2^2$$

and $\mathcal{D}$ a dictionary of 750 atoms randomly chosen in $\mathbb{R}^{500}$, where $A \in \mathbb{R}^{500 \times 500}$ and $b \in \mathbb{R}^{500}$ are also randomly chosen. We did not reduce $f$ to a closed-form expression simplifying computations. This is not a setting where BCG or CoGEnT can be applied. We depict two configurations of BMP: one with emphasis on sparsity of the iterates and one with emphasis on speed of convergence. The parameters $\eta_{\text{sparse}}$ and $\eta_{\text{fast}}$ were both optimized. We plot the results in Figure 13.

Figure 13: Comparison of BMP vs. GMP and OMP on $f : x \in \mathbb{R}^{500} \mapsto d_{\overline{\mathcal{B}}(0,1)}(Ax - b)^2$, with $\eta_{\text{sparse}} = 10$ and $\eta_{\text{fast}} = 2$.

We provide a sensitivity analysis of BMP to the parameter $\eta$ in Figure 14. The scaling is not exactly the same as in Figure 13 due to the randomness in the generation of the data. We see that $\eta \sim 2$ is an appropriate choice combining the best of speed and sparsity.

Figure 14: Sensitivity of BMP to the parameter $\eta$.

### A.6 Logistic regression

Here we compared BMP vs. GMP and OMP on the Gisette dataset [Guyon et al., 2005] available at https://archive.ics.uci.edu/ml/datasets/Gisette. We did not have access to the true parameters $x^*$ so we could not produce the NMSE plots. The objective function is the logistic loss

$$f : x \in \mathbb{R}^n \mapsto \frac{1}{m} \sum_{i=1}^{m} \ln(1 + \exp(-y_i a_i^\top x))$$

with the labels $y_i \in \{-1, 1\}$, and the dictionary is the set of signed canonical vectors $\mathcal{D} = \{\pm e_1, \dots, e_n\}$. We have $n = 5000$ and in order to reduce the running time, we chose $m = 1000$ and we slightly enhanced the code framework by replacing $\min_{v \in \mathcal{D}'} \langle \nabla f(x_t), v \rangle$ with $\min_{i \in [\![1,n]\!]} -|[\nabla f(x_t)]_i|$. Note that this lessens the speed-up provided by the weak-separation oracle in BMP. We represented the dual gaps of BMP by $\max_{i \in [\![1,n]\!]} |[\nabla f(x_t)]_i|$, like for GMP and OMP, and thus yielding a similar zig-zag plot.

Figure 15: Comparison of BMP vs. GMP and OMP on the Gisette dataset with $\eta_{\text{sparse}} = 3$ and $\eta_{\text{fast}} = 2$.

In this situation, Figure 15 shows that BMP converges as fast as GMP while producing iterates with much higher sparsity, equivalent to that of OMP. Hence, BMP hits the sweet spot of speed of convergence and sparsity of the iterates.

## B   Prerequisites for proofs

**Fact B.1.** *Let $f : \mathcal{H} \to \mathbb{R}$ be a coercive function and $(x_t)_{t \in \mathbb{N}}$ be a sequence of iterates in $\mathcal{H}$ such that $f(x_{t+1}) \leqslant f(x_t)$ for all $t \in \mathbb{N}$. Then $(x_t)_{t \in \mathbb{N}}$ is bounded.*

*Proof.* By assumption, $f(x_t) \leqslant f(x_0)$ for all $t \in \mathbb{N}$, so $\limsup_{t \to +\infty} f(x_t) \leqslant f(x_0) < +\infty$. Suppose $(x_t)_{t \in \mathbb{N}}$ is unbounded. Then there exists $\varphi : \mathbb{N} \to \mathbb{N}$ strictly increasing such that the subsequence $(x_{\varphi(t)})_{t \in \mathbb{N}}$ satisfies $\lim_{t \to +\infty} \|x_{\varphi(t)}\| = +\infty$. By coercivity, this implies that $\lim_{t \to +\infty} f(x_{\varphi(t)}) = +\infty$, and therefore $\limsup_{t \to +\infty} f(x_t) \geqslant +\infty$. This is absurd. $\qquad \square$

**Fact B.2.** *Let $f : \mathcal{H} \to \mathbb{R}$ be differentiable, $M > 0$, $\mu > 1$, and $x, v \in \mathcal{H}$ such that $\langle \nabla f(x), v \rangle \leqslant 0$. Define*

$$g : \gamma \in \mathbb{R}_+ \mapsto f(x) + \gamma \langle \nabla f(x), v \rangle + \frac{M}{\mu} \gamma^\mu \|v\|^\mu.$$

*Then*

$$\min_{\mathbb{R}_+} g = f(x) - \frac{\langle -\nabla f(x), v \rangle^{\overline{\mu}}}{\overline{\mu} M^{\overline{\mu}-1} \|v\|^{\overline{\mu}}}$$

*where* $\overline{\mu} := \mu/(\mu - 1) > 1$.

*Proof.* Let $\overline{\mu} = \mu/(\mu - 1)$. We have $\overline{\mu} - 1 = 1/(\mu - 1)$, and $g$ is differentiable with

$$\forall \gamma \in \mathbb{R}_+, \, g'(\gamma) \geqslant 0 \Leftrightarrow \langle \nabla f(x), v \rangle + M\gamma^{\mu-1}\|v\|^{\mu} \geqslant 0$$

$$\Leftrightarrow \gamma \geqslant \left( \frac{\langle -\nabla f(x), v \rangle}{M\|v\|^{\mu}} \right)^{1/(\mu-1)} = \frac{\langle -\nabla f(x), v \rangle^{\overline{\mu}-1}}{M^{\overline{\mu}-1}\|v\|^{\overline{\mu}}}.$$

Therefore, using $\mu(\overline{\mu} - 1) = \overline{\mu}$ and $1 - 1/\mu = 1/\overline{\mu}$,

$$\min_{\mathbb{R}_+} g = f(x) + \frac{\langle -\nabla f(x), v \rangle^{\overline{\mu}-1}}{M^{\overline{\mu}-1}\|v\|^{\overline{\mu}}} \nabla f(x)v + \frac{M}{\mu} \left( \frac{\langle -\nabla f(x), v \rangle^{\overline{\mu}-1}}{M^{\overline{\mu}-1}\|v\|^{\overline{\mu}}} \right)^{\mu} \|v\|^{\mu}$$

$$= f(x) - \frac{\langle -\nabla f(x), v \rangle^{\overline{\mu}}}{M^{\overline{\mu}-1}\|v\|^{\overline{\mu}}} + \frac{1}{\mu} \frac{\langle -\nabla f(x), v \rangle^{\mu(\overline{\mu}-1)}}{M^{\mu(\overline{\mu}-1)-1}\|v\|^{\mu(\overline{\mu}-1)}}$$

$$= f(x) - \left( 1 - \frac{1}{\mu} \right) \frac{\langle -\nabla f(x), v \rangle^{\overline{\mu}}}{M^{\overline{\mu}-1}\|v\|^{\overline{\mu}}}$$

$$= f(x) - \frac{\langle -\nabla f(x), v \rangle^{\overline{\mu}}}{\overline{\mu} M^{\overline{\mu}-1}\|v\|^{\overline{\mu}}}.$$

$\square$

**Corollary B.3.** *Let* $f : \mathcal{H} \to \mathbb{R}$ *be* $S$-*strongly convex of order* $s = 2$ *with* $\{x^*\} := \arg\min_{\mathcal{H}} f$. *Then for all* $x \in \mathcal{H}$,

$$f(x^*) \geqslant f(x) - \frac{\langle \nabla f(x), x - x^* \rangle^2}{2S\|x - x^*\|^2}.$$

*Proof.* Let $x \in \mathcal{H}$. By strong convexity, for all $\gamma \in \mathbb{R}_+$,

$$f(x + \gamma(x^* - x)) \geqslant f(x) + \gamma \langle \nabla f(x), x^* - x \rangle + \frac{S}{2}\gamma^2 \|x^* - x\|^2. \tag{3}$$

Let $v := x^* - x$, then $\langle \nabla f(x), v \rangle \leqslant 0$ by convexity. Applying Fact B.2 to the right-hand side of (3), since $\overline{s} = s/(s - 1) = 2$,

$$f(x + \gamma(x^* - x)) \geqslant f(x) - \frac{\langle \nabla f(x), x - x^* \rangle^2}{2S\|x - x^*\|^2}$$

so, with $\gamma = 1$,

$$f(x^*) \geqslant f(x) - \frac{\langle \nabla f(x), x - x^* \rangle^2}{2S\|x - x^*\|^2}.$$

$\square$

# C  The smooth strongly convex case

**Theorem C.1 (Smooth strongly convex case).** *Let* $\mathcal{D} \subset \mathcal{H}$ *be a dictionary such that* $0 \in \text{int}(\text{conv}(\mathcal{D}'))$ *and let* $f : \mathcal{H} \to \mathbb{R}$ *be* $L$-*smooth of order* $\ell = 2$ *and* $S$-*strongly convex of order* $s = 2$. *Then the Blended Matching Pursuit algorithm (Algorithm 3) ensures that* $f(x_t) - \min_{\mathcal{H}} f \leqslant \epsilon$ *for all* $t \geqslant T$ *where*

$$T = \mathcal{O}\left( \frac{L}{S} \ln \frac{|\phi_0|}{\epsilon} \right).$$

*Moreover,* $\|x_t - x^*\| \to 0$ *as* $t \to +\infty$ *at same rate.*

*Proof.* Let $\epsilon > 0$ and $T = N_{\text{dual}} + N_{\text{full}} + N_{\text{constrained}}$ where $N_{\text{dual}}$, $N_{\text{full}}$, and $N_{\text{constrained}}$ are the number of dual steps (Line 13), full steps (Line 17), and constrained steps (Line 7) taken in total respectively, and $\epsilon_t := f(x_t) - \min_{\mathcal{H}} f$. Similarly to Braun et al. [2017], we introduce *epoch starts* at iteration $t = 0$ or any iteration immediately following a dual step. Our goal is to bound the number of epochs and the number of iterations within each epoch. Notice that $0 \leqslant \epsilon_{t+1} \leqslant \epsilon_t$ and $\phi_t \leqslant \phi_{t+1} \leqslant 0$ for $t \in \mathbb{N}$.

Denote $\{x^*\} := \arg\min_{\mathcal{H}} f$. Let $t \in \mathbb{N}$ be an iteration of the algorithm and $v_t^{\text{FW}} \in \arg\min_{v \in \mathcal{D}'}\langle \nabla f(x_t), v \rangle = \arg\min_{z \in \text{conv}(\mathcal{D}')}\langle \nabla f(x_t), z \rangle$. We can assume that $f(x_t) > f(x^*)$ otherwise the iterates have already converged. By convexity, $\langle \nabla f(x_t), x^* - x_t \rangle < 0$. Since $0 \in \text{int}(\text{conv}(\mathcal{D}'))$, there exists $r > 0$ such that $\mathcal{B}(0, r) \subseteq \text{conv}(\mathcal{D}')$. Therefore, $\frac{r(x^* - x_t)}{2\|x^* - x_t\|} \in \text{conv}(\mathcal{D}')$ so

$$\min_{z \in \text{conv}(\mathcal{D}')}\langle \nabla f(x_t), z \rangle = \langle \nabla f(x_t), v_t^{\text{FW}} \rangle \leqslant \left\langle \nabla f(x_t), \frac{r(x^* - x_t)}{2\|x^* - x_t\|} \right\rangle < 0$$

and it follows that

$$\frac{\langle \nabla f(x_t), x_t - x^* \rangle}{\|x_t - x^*\|} \leqslant -\frac{2}{r}\langle \nabla f(x_t), v_t^{\text{FW}} \rangle. \tag{4}$$

By Corollary B.3,

$$f(x^*) \geqslant f(x_t) - \frac{\langle \nabla f(x_t), x_t - x^* \rangle^2}{2S\|x_t - x^*\|^2}.$$

Combining with (4) we obtain

$$\epsilon_t = f(x_t) - f(x^*) \leqslant \frac{2}{r^2 S}\langle \nabla f(x_t), v_t^{\text{FW}} \rangle^2. \tag{5}$$

Let $t$ be a dual step (Line 13). Then the weak-separation oracle call (Line 11) yields $\langle \nabla f(x_t), v_t^{\text{FW}} \rangle \geqslant \phi_t$. By (5) and Line 15,

$$\epsilon_t \leqslant \frac{2}{r^2 S}\phi_t^2 \tag{6}$$

$$= \frac{2}{r^2 S}\left(\frac{\phi_0}{\tau^{n_{\text{dual}}}}\right)^2 \tag{7}$$

where $n_{\text{dual}}$ is the number of dual steps taken before $t$. Therefore, by (7) and since $\tau > 1$, $N_{\text{dual}}$ is finite with

$$N_{\text{dual}} \leqslant \left\lceil \frac{1}{2}\log_\tau\left(\frac{2\phi_0^2}{r^2 S\epsilon}\right)\right\rceil. \tag{8}$$

If a full step is taken (Line 17), then the weak-separation oracle (Line 11) returns $v_t \in \mathcal{D}'$ such that $\langle \nabla f(x_t), v_t \rangle \leqslant \phi_t/\kappa$. By smoothness,

$$f(x_{t+1}) = \min_{\gamma \in \mathbb{R}} f(x_t + \gamma v_t)$$

$$\leqslant \min_{\gamma \in \mathbb{R}} f(x_t) + \gamma\langle \nabla f(x_t), v_t \rangle + \frac{L}{2}\gamma^2\|v_t\|^2$$

$$= f(x_t) - \frac{\langle \nabla f(x_t), v_t \rangle^2}{2L\|v_t\|^2}$$

$$\leqslant f(x_t) - \frac{(\phi_t/\kappa)^2}{2L(D_{\mathcal{D}'}/2)^2}$$

where we used $\|v_t\| \leqslant D_{\mathcal{D}'}/2$ (by symmetry). Therefore, the primal progress is at least

$$f(x_t) - f(x_{t+1}) \geqslant \frac{2\phi_t^2}{\kappa^2 L D_{\mathcal{D}'}^2}. \tag{9}$$

Last, if a constrained step is taken (Line 7), then by smoothness,

$$
\begin{aligned}
f(x_{t+1}) &= \min_{\gamma \in \mathbb{R}} f\big(x_t + \gamma \widetilde{\nabla} f(x_t)\big) \\
&\leqslant \min_{\gamma \in \mathbb{R}} f(x_t) + \gamma \langle \nabla f(x_t), \widetilde{\nabla} f(x_t) \rangle + \frac{L}{2}\gamma^2 \big\| \widetilde{\nabla} f(x_t) \big\|^2 \\
&= f(x_t) - \frac{\langle \nabla f(x_t), \widetilde{\nabla} f(x_t) \rangle^2}{2L \big\| \widetilde{\nabla} f(x_t) \big\|^2} \\
&= f(x_t) - \frac{\big\| \widetilde{\nabla} f(x_t) \big\|^2}{2L} \\
&\leqslant f(x_t) - \frac{\langle \widetilde{\nabla} f(x_t), v_t^{\text{FW-}\mathcal{S}} \rangle^2}{2L \big\| v_t^{\text{FW-}\mathcal{S}} \big\|^2} \\
&= f(x_t) - \frac{\langle \nabla f(x_t), v_t^{\text{FW-}\mathcal{S}} \rangle^2}{2L \big\| v_t^{\text{FW-}\mathcal{S}} \big\|^2} \\
&\leqslant f(x_t) - \frac{(\phi_t/\eta)^2}{2L(D_{\mathcal{D}'}/2)^2}
\end{aligned}
$$

where the last three lines respectively come from the Cauchy-Schwarz inequality, $v_t^{\text{FW-}\mathcal{S}} \in \text{span}(\mathcal{S}_t)$, $\langle \nabla f(x_t), v_t^{\text{FW-}\mathcal{S}} \rangle \leqslant \phi_t/\eta$ (Line 5), and $\big\| v_t^{\text{FW-}\mathcal{S}} \big\| \leqslant D_{\mathcal{D}'}/2$ (by symmetry). Therefore, the primal progress is at least

$$
f(x_t) - f(x_{t+1}) \geqslant \frac{2\phi_t^2}{\eta^2 L D_{\mathcal{D}'}^2}. \tag{10}
$$

whose lower bound only differs by a constant factor $(\kappa/\eta)^2$ from that of a full step (9).

Now, we have

$$
\begin{aligned}
T &= N_{\text{dual}} + N_{\text{full}} + N_{\text{constrained}} \\
&= N_{\text{dual}} + \sum_{\substack{t=0 \\ t \text{ epoch start}}}^{T-1} \left( N_{\text{full}}^{(t)} + N_{\text{constrained}}^{(t)} \right)
\end{aligned} \tag{11}
$$

where $N_{\text{full}}^{(t)}$ and $N_{\text{constrained}}^{(t)}$ are the number of full steps and constrained steps taken during epoch $t$ respectively. Let $t > 0$ be an epoch start. Thus, $t - 1$ is a dual step. By (6), since $x_t = x_{t-1}$ and $\phi_t = \phi_{t-1}/\tau$,

$$
\epsilon_t \leqslant \frac{2\tau^2 \phi_t^2}{r^2 S}. \tag{12}
$$

This also holds for $t = 0$ by (5) and Line 2 (and actually for all $t \in \mathbb{N}$). By (9) and (10), since $\phi_s = \phi_t$ for all nondual steps $s$ in the epoch starting at $t$,

$$
\begin{aligned}
\epsilon_t &\geqslant \sum_{s \in \text{epoch}(t)} \big( f(x_s) - f(x_{s+1}) \big) \\
&\geqslant \left( N_{\text{full}}^{(t)} + N_{\text{constrained}}^{(t)} \right) \frac{2\phi_t^2}{\max\{\kappa^2, \eta^2\} L D_{\mathcal{D}'}^2}.
\end{aligned} \tag{13}
$$

Combining (12) and (13),

$$
N_{\text{full}}^{(t)} + N_{\text{constrained}}^{(t)} \leqslant \frac{\tau^2 \max\{\kappa^2, \eta^2\} L D_{\mathcal{D}'}^2}{r^2 S}. \tag{14}
$$

Therefore, by (11) and (14),

$$
T \leqslant N_{\text{dual}} + (N_{\text{dual}} + 1) \frac{\tau^2 \max\{\kappa^2, \eta^2\} L D_{\mathcal{D}'}^2}{r^2 S}.
$$

By (8), we conclude that the algorithm converges with

$$T = \mathcal{O}\left(\frac{L}{S} \ln \frac{|\phi_0|}{\epsilon}\right).$$

Finally, by strong convexity and since $\nabla f(x^*) = 0$,

$$\frac{S}{2}\|x_t - x^*\|^2 \leqslant f(x_t) - f(x^*) - \langle \nabla f(x^*), x_t - x^* \rangle = \epsilon_t$$

for all $t \in \mathbb{N}$. Thus, $\|x_t - x^*\| \to 0$ as $t \to +\infty$. $\qquad\square$

## D  Main proofs

**Fact 2.1.** *Let $f : \mathcal{H} \to \mathbb{R}$ be smooth of order $\ell > 1$, convex, and sharp of order $\theta \in ]0, 1[$ on $\mathcal{K}$. Then $\theta \in ]0, 1/\ell]$.*

*Proof.* Let $x \in \mathcal{K} \setminus \arg\min_{\mathcal{H}} f$ and $x^* := \mathrm{proj}_{\arg\min_{\mathcal{H}} f}(x)$. By sharpness, smoothness, and $\nabla f(x^*) = 0$,

$$\mathrm{dist}\left(x, \arg\min_{\mathcal{H}} f\right) = \|x - x^*\| \leqslant C(f(x) - f(x^*))^\theta \leqslant C\left(\frac{L}{\ell}\right)^\theta \|x - x^*\|^{\ell\theta}.$$

Therefore,

$$\frac{1}{C}\left(\frac{\ell}{L}\right)^\theta \leqslant \|x - x^*\|^{\ell\theta - 1}.$$

As the left-hand side is constant and $x$ can be arbitrarily close to $x^*$, we conclude that $\ell\theta \leqslant 1$. $\qquad\square$

**Theorem 3.2 (Smooth convex case).** *Let $\mathcal{D} \subset \mathcal{H}$ be a dictionary such that $0 \in \mathrm{int}(\mathrm{conv}(\mathcal{D}'))$ and let $f : \mathcal{H} \to \mathbb{R}$ be smooth of order $\ell > 1$, convex, and coercive. Then the Blended Matching Pursuit algorithm (Algorithm 3) ensures that $f(x_t) - \min_{\mathcal{H}} f \leqslant \epsilon$ for all $t \geqslant T$ where $T = \mathcal{O}\big((L/\epsilon)^{1/(\ell-1)}\big)$.*

*Proof.* Let $\epsilon > 0$ and $T = N_{\mathrm{dual}} + N_{\mathrm{full}} + N_{\mathrm{constrained}} \in \mathbb{N} \cup \{+\infty\}$ where $N_{\mathrm{dual}}$, $N_{\mathrm{full}}$, and $N_{\mathrm{constrained}}$ are the number of dual steps (Line 13), full steps (Line 17), and constrained steps (Line 7) taken in total respectively. The objective $f$ is continuous and coercive so $\arg\min_{\mathcal{H}} f \neq \varnothing$. Let $\epsilon_t := f(x_t) - \min_{\mathcal{H}} f$ for $t \in \mathbb{N}$. Similarly to Braun et al. [2017], we introduce *epoch starts* at iteration $t = 0$ or any iteration immediately following a dual step. Our goal is to bound the number of epochs and the number of iterations within each epoch. Notice that $0 \leqslant \epsilon_{t+1} \leqslant \epsilon_t$ and $\phi_t \leqslant \phi_{t+1} \leqslant 0$ for $t \in \mathbb{N}$.

Let $x^* \in \arg\min_{\mathcal{H}} f$. The function $f$ is coercive and $f(x_{t+1}) \leqslant f(x_t)$ for $t \in \mathbb{N}$, so by Fact B.1 the sequence of iterates is bounded. Define $\rho := \sup_{t \in \mathbb{N}} \|x_t - x^*\| < +\infty$. Note that $\rho$ is independent of $T$. Let $t \in \mathbb{N}$ be an iteration of the algorithm, and $v_t^{\mathrm{FW}} \in \arg\min_{v \in \mathcal{D}'} \langle \nabla f(x_t), v \rangle = \arg\min_{z \in \mathrm{conv}(\mathcal{D}')} \langle \nabla f(x_t), z \rangle$. We can assume that $f(x_t) > f(x^*)$ otherwise the iterates have already converged. By convexity, $\langle \nabla f(x_t), x^* - x_t \rangle < 0$. Since $0 \in \mathrm{int}(\mathrm{conv}(\mathcal{D}'))$, there exists $r > 0$ such that $\mathcal{B}(0, r) \subseteq \mathrm{conv}(\mathcal{D}')$. Thus, $\frac{r(x^* - x_t)}{2\|x^* - x_t\|} \in \mathrm{conv}(\mathcal{D}')$ so

$$\min_{z \in \mathrm{conv}(\mathcal{D}')} \langle \nabla f(x_t), z \rangle = \langle \nabla f(x_t), v_t^{\mathrm{FW}} \rangle \leqslant \left\langle \nabla f(x_t), \frac{r(x^* - x_t)}{2\|x^* - x_t\|} \right\rangle < 0$$

i.e.,

$$\langle \nabla f(x_t), x_t - x^* \rangle \leqslant \frac{2\|x_t - x^*\|}{r} \left\langle -\nabla f(x_t), v_t^{\mathrm{FW}} \right\rangle$$

$$\leqslant \frac{2\rho}{r} \left\langle -\nabla f(x_t), v_t^{\mathrm{FW}} \right\rangle. \tag{15}$$

By convexity,

$$f(x_t) - f(x^*) \leqslant \langle \nabla f(x_t), x_t - x^* \rangle$$

so with (15),

$$\epsilon_t \leqslant \frac{2\rho}{r} \left\langle -\nabla f(x_t), v_t^{\text{FW}} \right\rangle. \tag{16}$$

Let $t$ be a dual step (Line 13). Then the weak-separation oracle call (Line 11) yields $\left\langle \nabla f(x_t), v_t^{\text{FW}} \right\rangle \geqslant \phi_t$. By (16) and Line 15,

$$\epsilon_t \leqslant \frac{2\rho}{r} |\phi_t| \tag{17}$$

$$= \frac{2\rho}{r} \frac{|\phi_0|}{\tau^{n_{\text{dual}}}} \tag{18}$$

where $n_{\text{dual}}$ is the number of dual steps taken before $t$. Therefore, by (18) and since $\tau > 1$,

$$N_{\text{dual}} \leqslant \left\lceil \log_\tau \left( \frac{2\rho|\phi_0|}{r\epsilon} \right) \right\rceil. \tag{19}$$

If a full step is taken (Line 17), then the weak-separation oracle (Line 11) returns $v_t \in \mathcal{D}'$ such that $\langle \nabla f(x_t), v_t \rangle \leqslant \phi_t/\kappa$. By smoothness and using Fact B.2 with $\bar\ell := \ell/(\ell-1) > 1$,

$$
\begin{aligned}
f(x_{t+1}) &\leqslant \min_{\gamma \in \mathbb{R}_+} f(x_t + \gamma v_t) \\
&\leqslant \min_{\gamma \in \mathbb{R}_+} f(x_t) + \gamma \langle \nabla f(x_t), v_t \rangle + \frac{L}{\ell} \gamma^\ell \|v_t\|^\ell \\
&= f(x_t) - \frac{\langle -\nabla f(x_t), v_t \rangle^{\bar\ell}}{\bar\ell L^{\bar\ell-1} \|v_t\|^{\bar\ell}} \\
&\leqslant f(x_t) - \frac{|\phi_t/\kappa|^{\bar\ell}}{\bar\ell L^{\bar\ell-1} (D_{\mathcal{D}'}/2)^{\bar\ell}}
\end{aligned}
$$

where we used $\|v_t\| \leqslant D_{\mathcal{D}'}/2$ (by symmetry). Therefore, the primal progress is at least

$$f(x_t) - f(x_{t+1}) \geqslant \frac{2^{\bar\ell} |\phi_t|^{\bar\ell}}{\bar\ell \kappa^{\bar\ell} L^{\bar\ell-1} D_{\mathcal{D}'}^{\bar\ell}}. \tag{20}$$

Last, if a constrained step is taken (Line 7), then by smoothness and using Fact B.2 with $-\widetilde\nabla f(x_t)$,

$$
\begin{aligned}
f(x_{t+1}) &\leqslant \min_{\gamma \in \mathbb{R}_+} f\big(x_t - \gamma \widetilde\nabla f(x_t)\big) \\
&\leqslant \min_{\gamma \in \mathbb{R}_+} f(x_t) - \gamma \langle \nabla f(x_t), \widetilde\nabla f(x_t) \rangle + \frac{L}{\ell} \gamma^\ell \big\| \widetilde\nabla f(x_t) \big\|^\ell \\
&= f(x_t) - \frac{\langle \nabla f(x_t), \widetilde\nabla f(x_t) \rangle^{\bar\ell}}{\bar\ell L^{\bar\ell-1} \big\| \widetilde\nabla f(x_t) \big\|^{\bar\ell}} \\
&= f(x_t) - \frac{\big\| \widetilde\nabla f(x_t) \big\|^{\bar\ell}}{\bar\ell L^{\bar\ell-1}} \\
&\leqslant f(x_t) - \frac{\big| \langle \widetilde\nabla f(x_t), v_t^{\text{FW-}\mathcal{S}} \rangle \big|^{\bar\ell}}{\bar\ell L^{\bar\ell-1} \big\| v_t^{\text{FW-}\mathcal{S}} \big\|^{\bar\ell}} \\
&\leqslant f(x_t) - \frac{\big| \langle \nabla f(x_t), v_t^{\text{FW-}\mathcal{S}} \rangle \big|^{\bar\ell}}{\bar\ell L^{\bar\ell-1} \big\| v_t^{\text{FW-}\mathcal{S}} \big\|^{\bar\ell}} \\
&\leqslant f(x_t) - \frac{|\phi_t/\eta|^{\bar\ell}}{\bar\ell L^{\bar\ell-1} (D_{\mathcal{D}'}/2)^{\bar\ell}}
\end{aligned}
$$

where the last three lines respectively come from the Cauchy-Schwarz inequality, $v_t^{\text{FW-}\mathcal{S}} \in \text{span}(\mathcal{S}_t)$, $\left\langle \nabla f(x_t), v_t^{\text{FW-}\mathcal{S}} \right\rangle \leqslant \phi_t/\eta$ (Line 5), and $\left\| v_t^{\text{FW-}\mathcal{S}} \right\| \leqslant D_{\mathcal{D}'}/2$ (by symmetry). Therefore, the primal progress is at least

$$f(x_t) - f(x_{t+1}) \geqslant \frac{2^{\bar{\ell}}|\phi_t|^{\bar{\ell}}}{\bar{\ell}\eta^{\bar{\ell}}L^{\bar{\ell}-1}D_{\mathcal{D}'}^{\bar{\ell}}} \tag{21}$$

whose lower bound only differs by a constant factor $(\kappa/\eta)^{\bar{\ell}}$ from that of a full step (20).

Now, we have

$$\begin{aligned}
T &= N_{\text{dual}} + N_{\text{full}} + N_{\text{constrained}} \\
&= N_{\text{dual}} + \sum_{\substack{t=0 \\ t \text{ epoch start}}}^{T-1} \left( N_{\text{full}}^{(t)} + N_{\text{constrained}}^{(t)} \right)
\end{aligned} \tag{22}$$

where $N_{\text{full}}^{(t)}$ and $N_{\text{constrained}}^{(t)}$ are the number of full steps and constrained steps taken during epoch $t$ respectively. Let $t > 0$ be an epoch start. Thus, $t - 1$ is a dual step. By (17), since $x_t = x_{t-1}$ and $\phi_t = \phi_{t-1}/\tau$,

$$\epsilon_t \leqslant \frac{2\rho\tau}{r}|\phi_t|. \tag{23}$$

This also holds for $t = 0$ by (16) and Line 2 (and actually for all $t \in [\![0, T]\!]$). By (20) and (21), since $\phi_s = \phi_t$ for all nondual steps $s$ in the epoch starting at $t$,

$$\begin{aligned}
\epsilon_t &\geqslant \sum_{s \in \text{epoch}(t)} \left( f(x_s) - f(x_{s+1}) \right) \\
&\geqslant \left( N_{\text{full}}^{(t)} + N_{\text{constrained}}^{(t)} \right) \frac{2^{\bar{\ell}}|\phi_t|^{\bar{\ell}}}{\bar{\ell}\max\{\kappa^{\bar{\ell}}, \eta^{\bar{\ell}}\}L^{\bar{\ell}-1}D_{\mathcal{D}'}^{\bar{\ell}}}
\end{aligned} \tag{24}$$

Combining (23) and (24),

$$N_{\text{full}}^{(t)} + N_{\text{constrained}}^{(t)} \leqslant \frac{2\rho\tau}{r} \frac{\bar{\ell}\max\{\kappa^{\bar{\ell}}, \eta^{\bar{\ell}}\}L^{\bar{\ell}-1}D_{\mathcal{D}'}^{\bar{\ell}}}{2^{\bar{\ell}}}|\phi_t|^{1-\bar{\ell}}. \tag{25}$$

Therefore, by (22), (25), and $\bar{\ell} > 1$,

$$\begin{aligned}
T &\leqslant N_{\text{dual}} + \frac{2\rho\tau}{r} \frac{\bar{\ell}\max\{\kappa^{\bar{\ell}}, \eta^{\bar{\ell}}\}L^{\bar{\ell}-1}D_{\mathcal{D}'}^{\bar{\ell}}}{2^{\bar{\ell}}} \sum_{t=0}^{N_{\text{dual}}} \left( \frac{|\phi_0|}{\tau^t} \right)^{1-\bar{\ell}} \\
&= N_{\text{dual}} + \frac{2\rho\tau}{r} \frac{\bar{\ell}\max\{\kappa^{\bar{\ell}}, \eta^{\bar{\ell}}\}L^{\bar{\ell}-1}D_{\mathcal{D}'}^{\bar{\ell}}}{2^{\bar{\ell}}}|\phi_0|^{1-\bar{\ell}} \frac{\tau^{(\bar{\ell}-1)(N_{\text{dual}}+1)} - 1}{\tau^{\bar{\ell}-1} - 1}.
\end{aligned}$$

By (19),

$$\begin{aligned}
T &\leqslant \log_\tau\left( \frac{2\rho|\phi_0|}{r\epsilon} \right) + 1 + \frac{2\rho\tau}{r} \frac{\bar{\ell}\max\{\kappa^{\bar{\ell}}, \eta^{\bar{\ell}}\}L^{\bar{\ell}-1}D_{\mathcal{D}'}^{\bar{\ell}}}{2^{\bar{\ell}}} \frac{|\phi_0|^{1-\bar{\ell}}}{\tau^{\bar{\ell}-1} - 1} \left( \tau^{(\bar{\ell}-1)\left(\log_\tau\left(\frac{2\rho|\phi_0|}{r\epsilon}\right)+2\right)} - 1 \right) \\
&= \log_\tau\left( \frac{2\rho|\phi_0|}{r\epsilon} \right) + 1 + \frac{2\rho\tau}{r} \frac{\bar{\ell}\max\{\kappa^{\bar{\ell}}, \eta^{\bar{\ell}}\}L^{\bar{\ell}-1}D_{\mathcal{D}'}^{\bar{\ell}}}{2^{\bar{\ell}}} \frac{|\phi_0|^{1-\bar{\ell}}}{\tau^{\bar{\ell}-1} - 1} \left( \tau^{2(\bar{\ell}-1)} \left( \frac{2\rho|\phi_0|}{r\epsilon} \right)^{\bar{\ell}-1} - 1 \right).
\end{aligned}$$

We conclude that the algorithm converges with

$$T = \mathcal{O}\left( \left( \frac{L}{\epsilon} \right)^{1/(\ell-1)} \right).$$

$\square$

**Theorem 3.3 (Smooth convex sharp case).** *Let $\mathcal{D} \subset \mathcal{H}$ be a dictionary such that $0 \in \mathrm{int}(\mathrm{conv}(\mathcal{D}'))$ and let $f : \mathcal{H} \to \mathbb{R}$ be $L$-smooth of order $\ell > 1$, convex, coercive, and $C$-sharp of order $\theta \in ]0, 1/\ell]$ on $\mathcal{K}$. Then the Blended Matching Pursuit algorithm (Algorithm 3) ensures that $f(x_t) - \min_\mathcal{H} f \leqslant \epsilon$ for all $t \geqslant T$ where*

$$T = \begin{cases} \mathcal{O}\left( C^{1/(1-\theta)} L^{1/(\ell-1)} \ln\left( \dfrac{C|\phi_0|}{\epsilon^{1-\theta}} \right) \right) & \text{if } \ell\theta = 1 \\ \mathcal{O}\left( \left( \dfrac{C^\ell L}{\epsilon^{1-\ell\theta}} \right)^{1/(\ell-1)} \right) & \text{if } \ell\theta < 1. \end{cases}$$

*Moreover, $\mathrm{dist}(x_t, \arg\min_\mathcal{H} f) \to 0$ as $t \to +\infty$ at same rate.*

*Proof.* Let $\epsilon > 0$. By Theorem 3.2, there exists $T \in \mathbb{N}$ such that $f(x_T) - \min_\mathcal{H} f \leqslant \epsilon$. Let $\epsilon_t := f(x_t) - \min_\mathcal{H} f$ for $t \in [\![0, T]\!]$ and $T = N_\text{dual} + N_\text{full} + N_\text{constrained}$ where $N_\text{dual}$, $N_\text{full}$, and $N_\text{constrained}$ are the number of dual steps (Line 13), full steps (Line 17), and constrained steps (Line 7) taken in total respectively. Similarly to Braun et al. [2017], we introduce *epoch starts* at iteration $t = 0$ or any iteration immediately following a dual step. Our goal is to bound the number of epochs and the number of iterations within each epoch. Notice that $0 \leqslant \epsilon_{t+1} \leqslant \epsilon_t$ and $\phi_t \leqslant \phi_{t+1} \leqslant 0$ for $t \in [\![0, T]\!]$.

Let $t \in [\![0, T]\!]$ be an iteration of the algorithm, $v_t^\text{FW} \in \arg\min_{v \in \mathcal{D}'} \langle \nabla f(x_t), v \rangle = \arg\min_{z \in \mathrm{conv}(\mathcal{D}')} \langle \nabla f(x_t), z \rangle$, and $x_t^* := \mathrm{proj}_{\arg\min_\mathcal{H} f}(x_t)$. We can assume that $f(x_t) > f(x_t^*)$ otherwise the iterates have already converged. By convexity, $\langle \nabla f(x_t), x_t^* - x_t \rangle < 0$. Since $0 \in \mathrm{int}(\mathrm{conv}(\mathcal{D}'))$, there exists $r > 0$ such that $\mathcal{B}(0, r) \subseteq \mathrm{conv}(\mathcal{D}')$. Therefore, $\frac{r(x_t^* - x_t)}{2\|x_t^* - x_t\|} \in \mathrm{conv}(\mathcal{D}')$ so

$$\min_{z \in \mathrm{conv}(\mathcal{D}')} \langle \nabla f(x_t), z \rangle = \langle \nabla f(x_t), v_t^\text{FW} \rangle \leqslant \left\langle \nabla f(x_t), \frac{r(x_t^* - x_t)}{2\|x_t^* - x_t\|} \right\rangle < 0$$

i.e.,

$$\frac{r\langle \nabla f(x_t), x_t - x_t^* \rangle}{-2\langle \nabla f(x_t), v_t^\text{FW} \rangle} \leqslant \|x_t - x_t^*\|. \tag{26}$$

The sharpness of $f$ implies that $\arg\min_\mathcal{H} f \subset \mathrm{int}(\mathcal{K})$. Let $r_t^* \in ]0, \|x_t - x_t^*\|[$ such that $\mathcal{B}(x_t^*, r_t^*) \subseteq \mathcal{K}$, and let $\rho := \min\{r_0^*/\|x_0 - x_0^*\|, \ldots, r_T^*/\|x_T - x_T^*\|\} \in ]0, 1[$. Then, $x_t^* + \rho(x_t - x_t^*) \in \mathcal{B}(x_t^*, r_t^*) \subseteq \mathcal{K}$. By convexity, $x_t^* = \mathrm{proj}_{\arg\min_\mathcal{H} f}(x_t^* + \rho(x_t - x_t^*))$: indeed, $\langle x_t^* - x^*, x_t - x_t^* \rangle \geqslant 0$ for all $x^* \in \arg\min_\mathcal{H} f$ by the Hilbert projection theorem, thus

$$\|(x_t^* + \rho(x_t - x_t^*)) - x^*\|^2 = \|x_t^* - x^*\|^2 + \rho^2\|x_t - x_t^*\|^2 + 2\rho\langle x_t^* - x^*, x_t - x_t^* \rangle$$
$$\geqslant \rho^2\|x_t - x_t^*\|^2 \tag{27}$$

where (27) is an equality if and only if $x^* = x_t^*$. Hence, using sharpness,

$$\rho\|x_t - x_t^*\| = \|(x_t^* + \rho(x_t - x_t^*)) - x_t^*\|$$
$$\leqslant C\big(f(x_t^* + \rho(x_t - x_t^*)) - f(x_t^*)\big)^\theta$$
$$\leqslant C\big(f(x_t^*) + \rho(f(x_t) - f(x_t^*)) - f(x_t^*)\big)^\theta$$
$$= C\rho^\theta(f(x_t) - f(x_t^*))^\theta \tag{28}$$
$$\leqslant C\rho^\theta\langle \nabla f(x_t), x_t - x_t^* \rangle^\theta$$

where the second and last inequalities come from convexity. Combining with (26), we get

$$\frac{r\langle \nabla f(x_t), x_t - x_t^* \rangle}{-2\langle \nabla f(x_t), v_t^\text{FW} \rangle} \leqslant \frac{C}{\rho^{1-\theta}}\langle \nabla f(x_t), x_t - x_t^* \rangle^\theta$$

so, by convexity, we obtain the primal bound

$$f(x_t) - f(x_t^*) \leqslant \langle \nabla f(x_t), x_t - x_t^* \rangle \leqslant \frac{1}{\rho}\left( -\frac{2C}{r}\langle \nabla f(x_t), v_t^\text{FW} \rangle \right)^{1/(1-\theta)}$$

i.e.,

$$\epsilon_t \leqslant \frac{1}{\rho}\left(-\frac{2C}{r}\left\langle \nabla f(x_t), v_t^{\mathrm{FW}}\right\rangle\right)^{1/(1-\theta)}. \tag{29}$$

Let $t$ be a dual step (Line 13). Then the weak-separation oracle call (Line 11) yields $\left\langle \nabla f(x_t), v_t^{\mathrm{FW}}\right\rangle \geqslant \phi_t$. By (29) and Line 15,

$$\epsilon_t \leqslant \frac{1}{\rho}\left(\frac{2C}{r}|\phi_t|\right)^{1/(1-\theta)} \tag{30}$$

$$= \frac{1}{\rho}\left(\frac{2C}{r}\frac{|\phi_0|}{\tau^{n_{\mathrm{dual}}}}\right)^{1/(1-\theta)} \tag{31}$$

where $n_{\mathrm{dual}}$ is the number of dual steps taken before $t$. Therefore, by (31) and since $\tau > 1$ and $\theta \in ]0, 1[$,

$$N_{\mathrm{dual}} \leqslant \left\lceil \log_\tau\left(\frac{2C|\phi_0|}{r\rho^{1-\theta}\epsilon^{1-\theta}}\right)\right\rceil. \tag{32}$$

If a full step is taken (Line 17), then the weak-separation oracle (Line 11) returns $v_t \in \mathcal{D}'$ such that $\langle \nabla f(x_t), v_t\rangle \leqslant \phi_t/\kappa$. By smoothness and using Fact B.2 and $\bar{\ell} := \ell/(\ell-1) > 1$,

$$
\begin{aligned}
f(x_{t+1}) &\leqslant \min_{\gamma\in\mathbb{R}_+} f(x_t + \gamma v_t)\\
&\leqslant \min_{\gamma\in\mathbb{R}_+} f(x_t) + \gamma\langle\nabla f(x_t), v_t\rangle + \frac{L}{\ell}\gamma^\ell\|v_t\|^\ell\\
&= f(x_t) - \frac{\langle-\nabla f(x_t), v_t\rangle^{\bar{\ell}}}{\bar{\ell}L^{\bar{\ell}-1}\|v_t\|^{\bar{\ell}}}\\
&\leqslant f(x_t) - \frac{|\phi_t/\kappa|^{\bar{\ell}}}{\bar{\ell}L^{\bar{\ell}-1}(D_{\mathcal{D}'}/2)^{\bar{\ell}}}
\end{aligned}
$$

where we used $\|v_t\| \leqslant D_{\mathcal{D}'}/2$ (by symmetry). Therefore, the primal progress is at least

$$f(x_t) - f(x_{t+1}) \geqslant \frac{2^{\bar{\ell}}|\phi_t|^{\bar{\ell}}}{\bar{\ell}\kappa^{\bar{\ell}}L^{\bar{\ell}-1}D_{\mathcal{D}'}^{\bar{\ell}}}. \tag{33}$$

Last, if a constrained step is taken (Line 7), then by smoothness and using Fact B.2,

$$
\begin{aligned}
f(x_{t+1}) &\leqslant \min_{\gamma\in\mathbb{R}_+} f\left(x_t - \gamma\widetilde{\nabla} f(x_t)\right)\\
&\leqslant \min_{\gamma\in\mathbb{R}_+} f(x_t) - \gamma\langle\nabla f(x_t), \widetilde{\nabla} f(x_t)\rangle + \frac{L}{\ell}\gamma^\ell\|\widetilde{\nabla} f(x_t)\|^\ell\\
&= f(x_t) - \frac{\langle\nabla f(x_t), \widetilde{\nabla} f(x_t)\rangle^{\bar{\ell}}}{\bar{\ell}L^{\bar{\ell}-1}\|\widetilde{\nabla} f(x_t)\|^{\bar{\ell}}}\\
&= f(x_t) - \frac{\|\widetilde{\nabla} f(x_t)\|^{\bar{\ell}}}{\bar{\ell}L^{\bar{\ell}-1}}\\
&\leqslant f(x_t) - \frac{\left|\langle\widetilde{\nabla} f(x_t), v_t^{\mathrm{FW\text{-}}\mathcal{S}}\rangle\right|^{\bar{\ell}}}{\bar{\ell}L^{\bar{\ell}-1}\|v_t^{\mathrm{FW\text{-}}\mathcal{S}}\|^{\bar{\ell}}}\\
&= f(x_t) - \frac{\left|\langle\nabla f(x_t), v_t^{\mathrm{FW\text{-}}\mathcal{S}}\rangle\right|^{\bar{\ell}}}{\bar{\ell}L^{\bar{\ell}-1}\|v_t^{\mathrm{FW\text{-}}\mathcal{S}}\|^{\bar{\ell}}}\\
&\leqslant f(x_t) - \frac{|\phi_t/\eta|^{\bar{\ell}}}{\bar{\ell}L^{\bar{\ell}-1}(D_{\mathcal{D}'}/2)^{\bar{\ell}}}
\end{aligned}
$$

where the last three lines respectively come from the Cauchy-Schwarz inequality, $v_t^{\text{FW-}\mathcal{S}} \in \text{span}(\mathcal{S}_t)$, $\langle \nabla f(x_t), v_t^{\text{FW-}\mathcal{S}} \rangle \leqslant \phi_t/\eta$ (Line 5), and $\|v_t^{\text{FW-}\mathcal{S}}\| \leqslant D_{\mathcal{D}'}/2$ (by symmetry). Therefore, the primal progress is at least

$$f(x_t) - f(x_{t+1}) \geqslant \frac{2^{\overline{\ell}}|\phi_t|^{\overline{\ell}}}{\overline{\ell}\eta^{\overline{\ell}}L^{\overline{\ell}-1}D_{\mathcal{D}'}^{\overline{\ell}}}. \tag{34}$$

whose lower bound only differs by a constant factor $(\kappa/\eta)^{\overline{\ell}}$ from that of a full step (33).

Now, we have

$$T = N_{\text{dual}} + N_{\text{full}} + N_{\text{constrained}}$$

$$= N_{\text{dual}} + \sum_{\substack{t=0 \\ t \text{ epoch start}}}^{T-1} \left( N_{\text{full}}^{(t)} + N_{\text{constrained}}^{(t)} \right) \tag{35}$$

where $N_{\text{full}}^{(t)}$ and $N_{\text{constrained}}^{(t)}$ are the number of full steps and constrained steps taken during epoch $t$ respectively. Let $t > 0$ be an epoch start. Thus, $t - 1$ is a dual step. By (30), since $x_t = x_{t-1}$ and $\phi_t = \phi_{t-1}/\tau$,

$$\epsilon_t \leqslant \frac{1}{\rho}\left( \frac{2\tau C}{r}|\phi_t| \right)^{1/1-\theta}. \tag{36}$$

This also holds for $t = 0$ by (29) and Line 2 (and actually for all $t \in [\![0, T]\!]$). By (33) and (34), since $\phi_s = \phi_t$ for all nondual steps $s$ in the epoch starting at $t$,

$$\epsilon_t \geqslant \sum_{s\in\text{epoch}(t)} \left( f(x_s) - f(x_{s+1}) \right)$$

$$\geqslant \left( N_{\text{full}}^{(t)} + N_{\text{constrained}}^{(t)} \right) \frac{2^{\overline{\ell}}|\phi_t|^{\overline{\ell}}}{\overline{\ell} \max\{\kappa^{\overline{\ell}}, \eta^{\overline{\ell}}\}L^{\overline{\ell}-1}D_{\mathcal{D}'}^{\overline{\ell}}}. \tag{37}$$

Combining (36) and (37),

$$N_{\text{full}}^{(t)} + N_{\text{constrained}}^{(t)} \leqslant \frac{1}{\rho}\left( \frac{2\tau C}{r} \right)^{1/(1-\theta)} \frac{\overline{\ell} \max\{\kappa^{\overline{\ell}}, \eta^{\overline{\ell}}\}L^{\overline{\ell}-1}D_{\mathcal{D}'}^{\overline{\ell}}}{2^{\overline{\ell}}}|\phi_t|^{1/(1-\theta)-\overline{\ell}}. \tag{38}$$

Therefore, by (35) and (38),

$$T \leqslant N_{\text{dual}} + \frac{1}{\rho}\left( \frac{2\tau C}{r} \right)^{1/(1-\theta)} \frac{\overline{\ell} \max\{\kappa^{\overline{\ell}}, \eta^{\overline{\ell}}\}L^{\overline{\ell}-1}D_{\mathcal{D}'}^{\overline{\ell}}}{2^{\overline{\ell}}} \sum_{t=0}^{N_{\text{dual}}} \left( \frac{|\phi_0|}{\tau^t} \right)^{1/(1-\theta)-\overline{\ell}} \tag{39}$$

$$= \begin{cases} N_{\text{dual}} + \frac{1}{\rho}\left( \frac{2\tau C}{r} \right)^{1/(1-\theta)} \frac{\overline{\ell} \max\{\kappa^{\overline{\ell}}, \eta^{\overline{\ell}}\}L^{\overline{\ell}-1}D_{\mathcal{D}'}^{\overline{\ell}}}{2^{\overline{\ell}}}(N_{\text{dual}} + 1) & \text{if } \ell\theta = 1 \\ N_{\text{dual}} + \frac{1}{\rho}\left( \frac{2\tau C}{r} \right)^{1/(1-\theta)} \frac{\overline{\ell} \max\{\kappa^{\overline{\ell}}, \eta^{\overline{\ell}}\}L^{\overline{\ell}-1}D_{\mathcal{D}'}^{\overline{\ell}}}{2^{\overline{\ell}}}|\phi_0|^{1/(1-\theta)-\overline{\ell}}\frac{\left( \tau^{\overline{\ell}-1/(1-\theta)} \right)^{N_{\text{dual}}+1} - 1}{\tau^{\overline{\ell}-1/(1-\theta)} - 1} & \text{if } \ell\theta < 1 \end{cases}$$

where, if $\alpha := \frac{1-\ell\theta}{(\ell-1)(1-\theta)} = \overline{\ell} - \frac{1}{1-\theta}$, by (32) we have

$$\left( \tau^{\overline{\ell}-1/(1-\theta)} \right)^{N_{\text{dual}}+1} = (\tau^{\alpha})^{N_{\text{dual}}+1}$$

$$= \exp\left( \alpha \ln(\tau)(N_{\text{dual}} + 1) \right)$$

$$\leqslant \exp\left( \alpha \ln(\tau) \left( \log_\tau\left( \frac{2C|\phi_0|}{r\rho^{1-\theta}\epsilon^{1-\theta}} \right) + 2 \right) \right)$$

$$= \exp\left( \alpha \ln\left( \frac{2C|\phi_0|}{r\rho^{1-\theta}\epsilon^{1-\theta}} \right) + 2\alpha \ln(\tau) \right)$$

$$= \tau^{2\alpha}\left( \frac{2C|\phi_0|}{r\rho^{1-\theta}\epsilon^{1-\theta}} \right)^\alpha.$$

By (32), we conclude that

$$
T = \begin{cases}
\mathcal{O}\left( C^{1/(1-\theta)} L^{1/(\ell-1)} \ln\left( \dfrac{C|\phi_0|}{\epsilon^{1-\theta}} \right) \right) & \text{if } \ell\theta = 1 \\[2em]
\mathcal{O}\left( \left( \dfrac{C^{\ell} L}{\epsilon^{1-\ell\theta}} \right)^{1/(\ell-1)} \right) & \text{if } \ell\theta < 1.
\end{cases}
$$

Finally, by (28),

$$
\|x_t - x_t^*\| \leqslant \frac{C}{\rho^{1-\theta}} \epsilon_t^{\theta}
$$

for all $t \in \mathbb{N}$. Thus, $\|x_t - x_t^*\| \to 0$ as $t \to +\infty$.  $\square$