[Reviews · NeurIPS 2019]

Reviewer 1



EDIT I was pleased with the authors answer about the duality gap. I encourage them to include this paragraph in their revision. I hope the public implementation of the code will be released for reproducibility. I have updated my grade from 6 to 7. ###################"" The article reads quite well, with numerous examples and explanations of convex optimization concepts. The methodology is strongly inspired from blended conditional gradient. The experimental validation is dubious to me as for loops are usually costly in python. The authors should at least use numba and @njit to use just in time compilation. How are the entries of X sampled ? If the design is near to orthogonal (eg if the entries are iid centered gaussian), then the problem is very easy and the figure may not reflect reality. Why not use real life datasets, eg from LIBSVM, and simulate only x^* and w (the observations y in these datasets could also be used)? I disagree with the sentence L15. Sparsity of the **solution** is important, but sparsity of the **iterates** does not affect generalization nor interpretation. Getting the sparsest solution is also important in the case when there are multiple ones, but that is not clear in this sentence. L35: which algorithm needs RIP? Usually RIP is needed to analyze the quality of the solution to Pb (1), not to analyze the convergence of an algorithm. Fact 2.1: please provide a reference. L83 I think some hypothesis is missing, f = indicator of {a} is strongly convex, but nabla f (x*) does not exist. The quantity \phi_t was not detailed enough in my opinion. How is it a dual gap estimate ? What is the reason for dividing it by tau at every dual step, and why are dual step coined like this? Finally, the code cannot be run on my machine: 363 364 # if still nothing --> 365 return _, 'FN', _ 366 367 def weak_sep(c, x, phi, kappa, S, D): NameError: name '_' is not defined And it seems that dual steps are never taken before: In [1]: %run bmp_code.py ********** η = 100 ********** Total CPU process time: 30.582956643000003 Number of constrained steps: 73 Number of dual steps: 0 Number of full steps: 39 ********** η = 10 ********** Total CPU process time: 30.691314241999997 Number of constrained steps: 41 Number of dual steps: 0 Number of full steps: 81 ********** η = 5 ********** Very cosmetic: L97 you have two different spelling for Lojasiewicz Algo 1 L5 L6 the authors may want to define \DeclareMathOperator{\argmin}{\mathrm{arg\,min}} to avoid the space appearing between arg and min because of the long text under. L52 can't it just be \lambda_j ? The notation H* L66/ R^n* L102 is confusing and I suggest the authors find a better one.

Reviewer 2



=== After the author's rebuttal, I decide to keep my score. The paper proposes a blended matching pursuit (BMP) algorithm, which combines weak-separation oracle, gradient descent and gradient matching steps to solve a smooth convex minimization problem restricted on a linear space spanned by a dictionary. Under various orders of smoothness and sharpness of the objective function, sublinear or linear convergence rates can be established for the proposed algorithm. Numerical experiments on synthetic data sets are conducted to show that BMP can achieve fast convergence and make solutions sparse. Overall, the work is interesting and important in applications with complete convergence discussion and supporting numerical results. However, the proposed algorithm seems the adaptation of the work by Braun et al. [2019] from a convex hull to a spanning linear space of a specified dictionary, which limits its novelty. In addition, computational experiments are insufficient to justify the claimed fast convergence, which requires more in-depth discussions, e.g., the influence of the matrix size $ A$, robustness to the noise, and sensitivity of sparsity levels. Other comments are shown as follows: 1. In the abstract, it claims that coordinate descent-like steps and stronger gradient descent steps which are unclear in the analysis of Algorithm 2. 2. In Problem (1), the domain for the variable $x$ should be changed to $span(\mathcal{D})$ according to the context. 3. In Lemma 2.3., the notation $(R^n)^*$ is a little confusing and could be replaced by another symbol. Likewise, $H^*$ could be confused with the dual space of $H$. 4. In line 231-232, is there any evidence to justify this statement? 5. In Section 3.1, strongly convex cases are skipped in convergence analysis, which should be commented at least. 6. Minor typos: In line 25, "lazified"? In line 40, "reader" -> "readers". In line 163, "Similarly" -> "Similar".

Reviewer 3



The main idea and the motivations are explained clearly. The experimental illustrations are convincing too. Some clarifications may help understanding the paper better. The dual gaps are not well defined within the main paper, which are being used in BMP to choose the steps. It makes more significant to discuss them in the main paper to appreciate the contrast better w.r.t. the referred paper (Braun et. al., 2019).

[Author Response · NeurIPS 2019]

First of all, we would like to thank the reviewers for their valuable feedback. Below we address each reviewer's
comments and place in **General** a discussion on the *dual gap estimates* $|\phi_t|$, as it was asked by multiple reviewers.

**[General]** The *duality gap* defined in the constrained setting (see, e.g., Jaggi [2013]) provides a good quality bound on
the primal gap and yields important insights for convergence analyses and algorithm design. In our setting, we can
derive a similar inequality as follows. Since $\mathcal{D}'$ is symmetric and $0 \in \mathrm{int}(\mathrm{conv}(\mathcal{D}'))$, there exists (an unknown) $\rho > 0$
such that $\{x^*, x_0, \ldots, x_T\} \subset \rho\,\mathrm{conv}(\mathcal{D}')$. Define $v_t^{\mathrm{FW}} := \mathrm{argmin}_{v \in \mathcal{D}'}\langle \nabla f(x_t), v\rangle$; note that $\langle \nabla f(x_t), v_t^{\mathrm{FW}}\rangle \leq 0$
since $\mathcal{D}' = -\mathcal{D}'$. Then $\epsilon_t := f(x_t) - f(x^*) \leq \langle \nabla f(x_t), x_t - x^*\rangle \leq \max_{(u,v) \in (\rho\,\mathrm{conv}(\mathcal{D}'))^2}\langle \nabla f(x_t), u - v\rangle =$
$-2\rho\langle \nabla f(x_t), v_t^{\mathrm{FW}}\rangle$ **(1)**, which is our desired inequality. The design of BMP strongly relies on this last quantity,
essential to the blending of full steps and constrained steps (see **(2)**) but unfortunately, it involves the unknown $\rho$ so it is
not functional here. We remedy this by working with *estimations* of these quantities: these are the *dual gap estimates*
$|\phi_t|$. We set $\phi_0 \leftarrow \langle \nabla f(x_0), v_0^{\mathrm{FW}}\rangle/\tau \leq 0$ (L2 in Alg. 3) so $f(x_0) - f(x^*) \leq 2\tau\rho|\phi_0|$ by **(1)**. Suppose $t'$ is the first
iteration where a *dual step* is taken. For all $t \in [\![0, t']\!]$, we have $\phi_t = \phi_0$ and $\epsilon_t \leq \epsilon_0$, so $\epsilon_t \leq 2\tau\rho|\phi_t|$. However, this
inequality becomes looser and looser, as the primal gaps $\epsilon_t$ decrease while the $|\phi_t|$ stay the same. Thus, when we
detect an improved dual bound via a negative weak-separation oracle call, i.e., $\langle \nabla f(x_t), v_t^{\mathrm{FW}}\rangle > \phi_t$ (see L11-12 in
Alg. 3 and the negative call in Oracle 2), the *dual step* updates the estimate $|\phi_t|$ accordingly: by **(1)**, this implies that
$\epsilon_t \leq 2\rho|\phi_t|$ so by setting $\phi_{t+1} \leftarrow \phi_t/\tau$ and $x_{t+1} \leftarrow x_t$, we obtain $\epsilon_{t+1} \leq 2\tau\rho|\phi_{t+1}|$. Therefore, a dual step updates
$|\phi_t|$ and tightens the bound on the primal gap (since $\tau > 1$). Furthermore, dividing by $\tau$ provides a geometric, hence
fast, rescaling of the dual gap estimates, which can also be seen in the proofs: the number of required dual steps is
$\mathcal{O}(\ln(1/\epsilon))$. In addition, note that $\phi_t$ is also an estimate for primal progress, roughly $\mathcal{O}(\phi_t^2)$ by smoothness; see **(2)**.

**(2)**: Assume WLOG that $f$ is $L$-smooth of order $\ell = 2$ and that the atoms have unit norm. It can be shown (see proofs)
that the progress $f(x_t) - f(x_{t+1})$ is at least $\langle \nabla f(x_t), v_t\rangle^2/2L$ for a full step, $\langle \nabla f(x_t), v_t^{\mathrm{FW}\text{-}\mathcal{S}}\rangle^2/2L$ for a constrained
step, and $\langle \nabla f(x_t), v_t^{\mathrm{FW}}\rangle^2/2L$ in GMP. Since $\phi_t$ is a (scaled) estimation of $\langle \nabla f(x_t), v_t^{\mathrm{FW}}\rangle$, the criterion L5 in Alg. 3
tests whether a constrained step would yield progress within a multiplicative factor to that of a GMP step, and this
without adding a new atom (hence preserving sparsity). The second test at L11-12 for deciding between a full step
(which is a GMP step with a weak-separation oracle) and a dual step is discussed above.

**[Reviewer #1] 1.** `for` loops: Indeed these are the typical example of poor Python performance. However, we compare
all methods using the same code framework and we suspect the results to differ only by a constant factor from those
using `numba`. We will provide evidence of this and modify the code to adapt to `numba`.
**2.** Data generation: We reproduced the CoGEnT experiments for comparison purposes, where the data is generated
using i.i.d. Gaussians and the design can be near orthogonal indeed. However we are in the process of conducting
additional experiments with more complex structures. These will be added in the revised version once all is finalized.
**3.** L15: Sparsity of the solution is indeed what is important however, in some applications, it is not necessarily the last
iterate: an earlier iterate might be selected as solution as it might provide better test error and avoid overfitting (early
stopping). Note that the lasso regularization or the $\ell_1$-ball constrained method also maintain sparsity at each iterate.
**4.** L35: Agreed, we mean that BMP does not require RIP or incoherence properties for its convergence to be analyzed.
**5.** Fact 2.1: Yes the cited paper [Nemirovskii and Nesterov, 1985] mentions such a result, we will provide the reference.
**6.** L83: Indeed, we will specify that we assume the function to be smooth.
**7.** About $\phi_t$: Please see **General**.
**8.** Code not working: We used underscores in the case where some values can be ignored when unpacking, e.g., `a,`
`_, c = (1, 2, 3)`, and this raised an error on your machine. We prepared an online version at https://colab.
research.google.com/drive/1VpVET2_lw6kEXttRabIfpHdFiwo4Hp1V. Apart from Figure 2, each experiment
takes an average total ∼15 minutes to run. The time limit can be reduced by setting `time_tol` (in seconds) to a lower
value; `time_tol` is set at the beginning of each experiment.
**9.** L97: Indeed the correct spelling is Łojasiewicz. L152: Yes we could use $\lambda_{i_j}$ since the correction option L22 is never
used in our experiments. L66: We will remove the $\mathcal{H}^*$ notation and simply use $\mathrm{argmin}_{\mathcal{H}} f$.

**[Reviewer #2] 1.** The coordinate-like steps correspond to the *full steps* $x_{t+1} \leftarrow x_t + \gamma_t v_t$ (L17), where a step is taken
in the direction of an atom. This reduces to a coordinate step if the atoms are the canonical vectors (in finite dimension)
$\pm e_1, \ldots, \pm e_n$. The gradient descent steps are the *constrained steps* $x_{t+1} \leftarrow x_t - \gamma_t \widetilde{\nabla} f(x_t)$ (L7).
**2.** Actually $\mathrm{span}(\mathcal{D}) = \mathcal{H}$ by definition of $\mathcal{D}$. There is no need to address the case $\mathrm{span}(\mathcal{D}) \subsetneq \mathcal{H}$ as a simple reduction
$\mathcal{H} \leftarrow \mathrm{span}(\mathcal{D})$ would yield the same analyses.
**3.** Yes we will remove the $\mathcal{H}^*$ notation and simply use $\mathrm{argmin}_{\mathcal{H}} f$.
**4.** We have no evidence of this and it is an open problem we are interested in.
**5.** Indeed we could comment, e.g., that the rate derived is the same as that of the sharp case, up to a constant factor.
Note however that the sharp case subsumes the strongly convex case as mentioned L83-85, L95-96, and L221-223.
**6.** The term "lazified" comes from the cited paper [Braun et al., 2017].

**[Reviewer #3]** Please see **General**.

[Meta-Review · NeurIPS 2019]

This paper proposes a new algorithm for minimizing convex objective functions. The reviewers found the article clearly written and the material presented could be useful to the broader ML community. The authors provided satisfying explanations and improvements in their response. There were some questions remaining about the simplicity of the simulation scenarios, but the authors have promised to make more realistic simulation experiments available. This paper seems to be above the bar of acceptance because of the theoretical and empirical evidence connected to this new algorithm.